# PDE-SSM: A Spectral State Space Approach to Spatial Mixing in Diffusion Transformers

## Abstract

The success of vision transformers—especially for generative modeling—is limited by the quadratic cost and weak spatial inductive bias of self-attention. We propose PDE-SSM, a spatial state-space block that replaces attention with a learnable convection–diffusion–reaction partial differential equation. This operator encodes a strong spatial prior by modeling information flow via physically grounded dynamics rather than all-to-all token interactions. Solving the PDE in the Fourier domain yields global coupling with near-linear complexity of $O(N \log N)$, delivering a principled and scalable alternative to attention. We integrate PDE-SSM into a flow-matching generative model to obtain the PDE-based Diffusion Transformer PDE-SSM-DiT. Empirically, PDE-SSM-DiT matches or exceeds the performance of state-of-the-art Diffusion Transformers while substantially reducing compute. Our results show that, analogous to 1D settings where SSMs supplant attention, multi-dimensional PDE operators provide an efficient, inductive-bias-rich foundation for next-generation vision models.

## 1 Introduction

The Transformer architecture (Vaswani et al., 2017), with its self-attention mechanism, has become the de facto standard for state-of-the-art machine learning techniques, extending its dominance from Natural Language Processing (Achiam et al., 2023; Touvron et al., 2023) to Computer Vision (Dosovitskiy et al., 2021; Peebles and Xie, 2023). In generative modeling for images, the U-Net (Ronneberger et al., 2015) backbone with attention mechanism has become the standard for diffusion models over the years (Ho et al., 2020; Song et al., 2020; Nichol and Dhariwal, 2021; Dhariwal and Nichol, 2021; Rombach et al., 2022; Lipman et al., 2023). However, after a long streak of success, the Diffusion Transformer (DiT) (Peebles and Xie, 2023) marked a significant milestone by replacing the U-Net backbone with a Transformer-based architecture, demonstrating that pure Transformer models can achieve or surpass the performance of convolutional U-Net backbones in large-scale diffusion models. Like other transformers, DiTs utilize deep networks. However, it is worth noting that DiTs are also constructed using attention mechanisms, which makes them fundamentally scalable. Be it modern U-Nets or DiTs, these are attention-based models which essentially operate by treating an image as a collection of patches and apply full weight within the patch and self-attention to model the relationship between patches (Dosovitskiy et al., 2021; Liu et al., 2021; Peebles and Xie, 2023).

Despite its empirical success, self-attention has two well-known drawbacks in the spatial domain Tay et al. (2022), which are the focus of this paper. The limitations of self-attention in this case are as follows: *first*, its computational and memory requirements become increasingly challenging in 2D inputs, which have quadratically as many tokens as standard 1D inputs. Although there are a few recent ideas to attempt and reduce this cost (e.g. linear attention) Katharopoulos et al. (2020); Choromanski et al. (2020); Shen et al. (2018), their performance generally falls short compared to standard attention-based methods. *Second*, self-attention is fundamentally permutation-invariant, meaning it does not inherently encode spatial locality or grid structure. Positional encodings are commonly introduced to remedy this limitation, and indeed, they allow Transformers to capture ordering and relative spatial information. Nevertheless, this mechanism is not as direct or specialized as the inductive bias present in convolutional neural networks (CNNs), which are explicitly designed around locality. Consequently, Transformer-based models often rely more heavily on data to learn robust spatial relationships.

Concurrently, a distinct lineage of sequence models has emerged to address these limitations in the 1D domain. State Space Models (SSMs), particularly structured SSMs (S4) (Gu et al., 2022) and their modern successors like Mamba (Gu and Dao, 2023), have achieved remarkable success on long-sequence tasks while offering compelling efficiency. These models are grounded in the theory of continuous-time systems, governed by a linear Ordinary Differential Equation (ODE):

$$\frac{dh(t)}{dt} = Ah(t) + Bu(t). \tag{1}$$

By imposing structure on the state matrix $A$, these models can be formulated as a recurrent or convolutional system, enabling them to capture extremely long-range dependencies in linear or near-linear time (Gu et al., 2022). Their success raises a critical and compelling question: *how can the principles of SSMs be generalized from 1D time to N-D space to create a more efficient and spatially-aware foundation for vision models?*

Recent work has extended SSMs to graphs (Lahoti et al., 2025), replacing $A$ with the adjacency matrix. In the context of SSMs in 2D, initial forays into this area, such as Vision Mamba (ViM) (Zhu et al., 2024), have adapted 1D SSMs to images by "flattening" the 2D image domain into a 1D sequence, followed by applying the SSM model along forward and backward paths. While effective, this approach imposes an artificial causality on the spatial domain, effectively processing pixels/nodes in a scan order rather than respecting the true multi-dimensional structure of images. In this sense, it represents a heuristic adaptation of 1D principles, rather than a full generalization to the spatial setting. We provide an additional discussion of concurrent and complementary approaches in Appendix A.

**Our approach.** To bridge this gap, in this work, we propose a more principled approach. Concretely, we argue that the natural continuous-space generalization of an ODE that is the core of SSMs as shown in Equation (1), is a Partial Differential Equation (PDE). This observation has been studied for the control of PDEs in the context of chemical engineering Morris and Levine (2010); Foias et al. (1996). We extend this idea and introduce the *PDE State-Space Model (PDE-SSM)*, an architectural block that models the evolution of a hidden state $h(t, \mathbf{x})$ in time $t$, where $\mathbf{x} \in \mathbb{R}^d$ — according to a learnable, general-form diffusion–convection–reaction equation, as follows:

$$\frac{\partial h}{\partial t} = \underbrace{\nabla \cdot (K \nabla h)}_{\text{Diffusion}} + \underbrace{\mathbf{b} \cdot \nabla h}_{\text{Convection}} + \underbrace{rh}_{\text{Reaction}}, \tag{2}$$

where the diffusion tensor $K$, convection vector field $\mathbf{b}$, and reaction term $r$ are learnable parameters. This formulation provides a flexible and physically-grounded inductive bias; information propagates across the spatial domain via structured and well-studied mechanisms: (i) diffusion smooths and aggregates local information; (ii) convection directs the flow of information; and (iii) reaction models local feature transformations. Importantly, by implementing the solution to the PDE in Equation (2) in the Fourier domain, the underlying operator admits a computational complexity of $O(N \log N)$, making it highly scalable relative to the $O(N^2)$ cost of self-attention, while still capturing global interactions more directly than local convolutions. We integrate the PDE-SSM block into a standard transformer architecture, replacing the self-attention layers to create the PDE-based Diffusion Transformer, to obtain PDE-SSM$_{\text{DiT}}$.

**Our contributions are as follows:**

1. We introduce PDE-SSM, a novel architectural block that provides a principled generalization of State Space Models to multi-dimensional spatial data by leveraging learnable PDEs.

2. We present PDE-SSM$_{\text{DiT}}$, a new backbone for generative vision models that utilizes the PDE-SSM for efficient and spatially-aware feature mixing, efficiently implemented via the Fourier transform.

3. Through extensive experiments on both low- and high-resolution image generation tasks, we show that PDE-SSM$_{\text{DiT}}$ is an effective generative model, achieving competitive or superior performance to the standard DiT, while being more computationally efficient, especially at higher resolutions.

## 2 FROM 1D SSM TO A SPATIAL PDE-SSM

In this section, we develop the PDE-SSM as a principled generalization of State Space Models from 1D sequences to multi-dimensional spatial data. We begin by reformulating the 1D SSM from a differential operator perspective in Section 2.1, which provides a natural bridge to our spatial, 2D formulation in Section 2.2, followed by its efficient implementation in Section 2.3.

### 2.1 A DIFFERENTIAL OPERATOR VIEW OF SSMS

We start by utilizing the known connection between SSMs and ODEs, and their solutions (Evans, 2022): a linear time-invariant SSM (Equation (1)), can be rewritten as the solution to an ODE:

$$\mathcal{L}h(t) = Bu(t), \quad \text{where} \quad \mathcal{L} = \frac{d}{dt} - A. \tag{3}$$

Here, $u(t)$ is the input signal, $h(t)$ is the hidden state at time $t$, and $A$, $B$ are learnable matrices. $\mathcal{L}$ is a first-order linear differential operator that acts on the hidden state $h(t)$. The solution can be expressed as a convolution with the Green's function (or impulse response) of the operator $\mathcal{L}$:

$$h(t) = (\mathcal{G}_A \star Bu)(t), \quad \text{where} \quad \mathcal{G}_A(t) = e^{tA}\mathbb{I}(t \geq 0). \tag{4}$$

The computation described in Equation (4) is *convolutional* — which is key to the efficiency of modern SSMs (Gu et al., 2022). By structuring the matrix $A$, the convolutional kernel $\mathcal{G}_A$ can be computed efficiently, in some cases in the Fourier domain Gu et al. (2022), allowing the model to capture long-range dependencies in near-linear time.

### 2.2 GENERALIZING TO SPACE: THE PDE-SSM FORMULATION

To extend the SSM framework from a 1D sequence, temporal domain to a multi-dimensional spatial domain $\mathbf{x} \in \mathbb{R}^{n_d}$, we replace the ODE with a PDE. This is a natural generalization, because PDEs are the fundamental mathematical framework for describing systems that evolve over both time and space. The resulting operator, which we call *PDE-SSM*, maps an input feature map $u(\mathbf{x})$ to an output feature map $h(\mathbf{x})$ via a spatial convolution, as follows:

$$h(\mathbf{x}) = (\mathcal{G}_\zeta \star \mathcal{B}_\gamma u)(\mathbf{x}). \tag{5}$$

The PDE-SSM operation, described in Equation (5), consists of two stages: (i) an input embedding $\mathcal{B}_\gamma u$ to obtain the initial state; and (ii) a spatial evolution defined by the Green's function $\mathcal{G}_\zeta$ of our learnable PDE. In what follows, we formalize these two operations.

**The Embedding Operator $\mathcal{B}_\gamma$.** The operator $\mathcal{B}_\gamma$ is a learnable embedding layer that transforms the input $u$ into the hidden space. For a 2D image, which is at the focus of this paper, it reads:

$$\mathcal{B}_\gamma u = (\gamma_0 I + \gamma_x \frac{\partial}{\partial x} + \gamma_y \frac{\partial}{\partial y})(R \star u)(\mathbf{x}). \tag{6}$$

Here, $R$ is a $1 \times 1$ (also known as pointwise) 2D convolution that mixes the input channels of $u$. The learnable scalars $\gamma_0, \gamma_x, \gamma_y$ then create a new feature map from a weighted sum of the channel-mixed features and their spatial gradients $\frac{\partial}{\partial x}$ and $\frac{\partial}{\partial y}$.

**The PDE Evolution and its Green's Function $\mathcal{G}_\zeta$.** The core of our model is the diffusion-convection-reaction PDE from Equation (2), which governs the evolution of the state $h(\mathbf{x}, t)$ from an initial condition $h(\mathbf{x}, t = 0) = \mathcal{B}_\gamma u(\mathbf{x})$. The PDE-SSM state is obtained by solving the PDE Equation 2 in the Fourier domain to derive an analytic expression for its Green's function. To this end, let $\hat{h}(\mathbf{k}, t)$ be the spatial Fourier transform of $h(\mathbf{x}, t)$, where $\mathbf{k} = (k_x, k_y)$ is the corresponding frequency vector. In Fourier space, the spatial derivatives $\frac{\partial}{\partial x}$ and $\frac{\partial}{\partial y}$ transform into multiplication by $i\mathbf{k}$: $\nabla \to i\mathbf{k}$. Overall, applying the Fourier transform to the PDE yields:

$$\frac{\partial \hat{h}(\mathbf{k}, t)}{\partial t} = (i\mathbf{k})^T K(i\mathbf{k})\hat{h}(\mathbf{k}, t) + \mathbf{b} \cdot (i\mathbf{k})\hat{h}(\mathbf{k}, t) + r\hat{h}(\mathbf{k}, t) \tag{7}$$

$$= \underbrace{(-\mathbf{k}^T K\mathbf{k} + r + i(\mathbf{b} \cdot \mathbf{k}))}_{\lambda(\mathbf{k})} \hat{h}(\mathbf{k}, t). \tag{8}$$

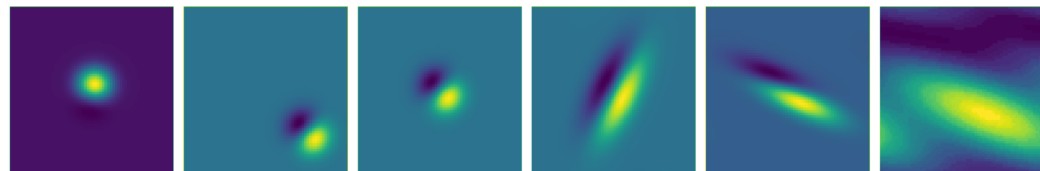

Figure 1: **Visualizing the PDE-SSM Convolutional Kernels.** By sampling the learnable parameters $\xi = (\mathcal{B}_\gamma, \zeta)$, our PDE-SSM can represent a diverse family of convolutional kernels. The examples show kernels that are (from left to right): localized, directionally blurred (anisotropic diffusion), shifted (convection), and a combination of effects. This flexibility allows our PDE-SSM model to learn a rich basis for spatial feature mixing, including non-local connections.

This transforms the PDE into a simple, linear ODE for each frequency $\mathbf{k}$, with the solution:

$$\hat{h}(\mathbf{k}, \tau) = e^{\tau \lambda(\mathbf{k})} \hat{h}(\mathbf{k}, 0) \tag{9}$$

The term $e^{\tau \lambda(\mathbf{k})}$ is the Fourier transform of the Green's function, $\hat{\mathcal{G}}_\zeta(\mathbf{k})$, also known as the symbol of the operator, and the kernel is defined in the frequency domain by the parameters $\zeta = (K, \mathbf{b}, r, \tau)$:

$$\hat{\mathcal{G}}_\zeta(\mathbf{k}) = \exp\left(\tau(-\mathbf{k}^T K \mathbf{k} + r + i(\mathbf{b} \cdot \mathbf{k}))\right). \tag{10}$$

**Theoretical Properties of PDE-SSM.** Beyond computational efficiency, the PDE-SSM operator inherits structural guarantees from its PDE formulation. Stability follows naturally: since the diffusion tensor $K$ is constrained to be positive semidefinite, the real part of the spectrum $\lambda(\mathbf{k})$ in Eq. (8) is non-positive, ensuring that the evolution operator cannot introduce uncontrolled growth across frequencies—a property not inherent to standard attention. Expressivity is preserved through the combination of diffusion, convection, and reaction terms, which together span a rich class of kernels that includes localized filters, global translations, and nonlocal smoothers, as illustrated in Figure 1. Nonlocality is inherent as well: the integration time $\tau$ directly controls the receptive field, and as $\tau$ increases, the operator aggregates information across arbitrarily large spatial scales, paralleling the long-range memory of 1D SSMs but in the spatial domain. These guarantees make PDE-SSM a principled and theoretically grounded alternative to attention for spatial mixing.

The closed-form expression in Eq. (10) enables efficient computation while allowing the learnable parameters to encode physically meaningful inductive biases:

- **Diffusion ($K$):** The term $-\tau \mathbf{k}^\top K \mathbf{k}$ is a non-positive quadratic form, acting as a low-pass filter that damps high frequencies. Anisotropic $K$ induces direction-dependent smoothing.

- **Convection (b):** The term $i\tau(\mathbf{b} \cdot \mathbf{k})$ induces a phase shift $e^{i\tau(\mathbf{b} \cdot \mathbf{k})}$, corresponding to a spatial translation via the Fourier shift theorem, enabling directed nonlocal relationships.

- **Reaction ($r$):** The scalar $\tau r$ uniformly scales responses across all frequencies, modeling global amplification ($r > 0$) or suppression ($r < 0$).

- **Time ($\tau$):** The integration time modulates the strength of all effects: as $\tau \to 0$, the operator approaches the identity, while larger $\tau$ values yield increasingly global kernels.

We illustrate different kernels obtained by our PDE-SSM convolution kernel in Equation (10) in Figure 1. We note that, in the above discussion, we have presented the computation for a single channel. In practice, one needs to consider the multi-channel case. Let $C$ be the number of channels. In this case, in 2D, the matrix $K$ becomes a 4D tensor of size $C \times C \times 2 \times 2$, the vector $\mathbf{b}$ becomes a $C \times C \times 2 \times 1$ tensor and $r$ becomes a $C \times C \times 1 \times 1$ tensor. The convolution is performed for each channel pair, and the resulting outputs are aggregated across channels, similar to the way that standard multi-channel 2D convolutions are implemented in modern deep learning frameworks.

### 2.3 Efficient Implementation with Multi-Channel Coupling

The analytic form of the kernel in the Fourier domain in Equation (10) allows for an efficient implementation. To model rich interactions where channels interact, i.e., mix, we generalize the PDE-SSM parameters $\zeta = (K, \mathbf{b}, r, \tau)$ from scalars to matrices that operate on the channel dimension.

---

### Algorithm 1: **PDE-SSM Forward Pass with Channel Coupling**

**Require:** Input tensor $\mathbf{u} \in \mathbb{R}^{B \times C_{\text{in}} \times H \times W}$.
**Require:** Learnable parameters $\mathbf{R} \in \mathbb{R}^{C_{\text{hid}} \times C_{\text{in}}}, \boldsymbol{\Gamma}_0, \boldsymbol{\Gamma}_x, \boldsymbol{\Gamma}_y, \mathbf{K}_{xx}, \ldots, \in \mathbb{R}^{C_{\text{hid}} \times C_{\text{hid}}}, \tau$.

1   **1. Input Embedding in Fourier Domain**

2   $\mathbf{u}' \leftarrow \text{Conv}_{1 \times 1}(\mathbf{u}, \mathbf{R})$            ▷ Project to $C_{\text{hid}}$ channels

3   $\hat{\mathbf{u}}' \leftarrow \text{FFT}(\mathbf{u}')$                ▷ Transform to frequency domain

4   Generate $\mathbf{k} = (k_x, k_y)$ as the discrete frequency grid.

5   $\hat{\mathcal{B}}(\mathbf{k}) \leftarrow \boldsymbol{\Gamma}_0 + ik_x \boldsymbol{\Gamma}_x + ik_y \boldsymbol{\Gamma}_y$          ▷ Matrix symbol for $\mathcal{B}$

6   $\hat{\mathbf{v}}(\mathbf{k}) \leftarrow \hat{\mathcal{B}}(\mathbf{k}) \cdot \hat{\mathbf{u}}'(\mathbf{k})$       ▷ Apply embedding (matrix–vector product)

7   **2. Coupled PDE Evolution in Fourier Domain**

8   $\boldsymbol{\Lambda}(\mathbf{k}) \leftarrow -(\mathbf{k}^\top \mathbf{K} \mathbf{k}) + \mathbf{R}_m + i(\mathbf{k} \cdot \mathbf{B})$     ▷ Assemble $C_{\text{hid}} \times C_{\text{hid}}$ matrix for each $\mathbf{k}$

9   $\hat{\mathcal{G}}_\zeta(\mathbf{k}) \leftarrow \exp(\tau \boldsymbol{\Lambda}(\mathbf{k}))$           ▷ Green's function symbol

10   $\hat{\mathbf{h}}(\mathbf{k}) \leftarrow \hat{\mathcal{G}}_\zeta(\mathbf{k}) \cdot \hat{\mathbf{v}}(\mathbf{k})$      ▷ Apply PDE operator (channelwise matrix–vector)

11   **3. Return to Spatial Domain**

12   $\mathbf{h} \leftarrow \text{iFFT}(\hat{\mathbf{h}})$                 ▷ Back to spatial domain

13   **return h**

---

This extension transforms the decoupled $C$ PDE-SSM systems into a system of coupled PDEs, which can be solved simultaneously in the Fourier domain. The procedure is detailed in Algorithm 1. In this formulation, the parameters for embedding ($\boldsymbol{\Gamma}$'s) and evolution ($\mathbf{K}, \mathbf{B}, \mathbf{R}_m$) are now $C_{\text{hid}} \times C_{\text{hid}}$ matrices. Consequently, the operations in lines 6, 8, 9, and 10 in Algorithm 1 become matrix-matrix operations on the channel direction.

**Computational Complexity.** Channel-coupling impacts the computational cost. While the FFTs remain $\mathcal{O}(C_{\text{hid}} \cdot N \log N)$, the Fourier-domain operations have the following complexity: *Symbol Computation (Lines 5, 8, Algorithm 1)*—assembling the $\boldsymbol{\Lambda}(\mathbf{k})$ matrix for each of the $N$ frequencies involves matrix additions and scalar–matrix products, costing $\mathcal{O}(N \cdot C_{\text{hid}}^2)$; *Matrix–Vector Products (Lines 6, 10, Algorithm 1)*—these cost $\mathcal{O}(N \cdot C_{\text{hid}}^2)$. The total complexity is therefore dominated by the matrix–matrix products, leading to an overall cost of $\mathcal{O}(C_{\text{hid}} \cdot N \log N + N \cdot C_{\text{hid}}^2)$. The $N \cdot C_{\text{hid}}^2$ term dominates when the number of hidden channels is large.

## 3   USING PDE-SSM WITHIN A DIFFUSION TRANSFORMER

We now discuss the viability of the PDE-SSM block in the context of Diffusion Transformer (DiT), to address the task of image generation (Lipman et al., 2023; Kingma et al., 2024; Song et al., 2021).

**Background on Flow-Matching.** We focus our attention on flow-matching techniques as discussed in Lipman et al. (2023), where the goal is to train a transformer-based network that solves the following optimization problem:

$$\min_\theta \mathbb{E}_{u,z} \int_0^1 \|v_\theta(tu + (1-t)z, t) - (u - z)\|^2 \, dt, \tag{11}$$

where $u$ is the original ("clean") image, $z$ is Gaussian noise and $t$ is time. During training, we are given a corrupt, noisy image

$$u_t = tu + (1-t)z, \tag{12}$$

and the goal is to train a network $v_\theta$ that recovers the velocity $u - z$. This task is closely related to image denoising, because given $v_\theta$, one can estimate the clean image $u$ by setting:

$$u \approx u_t + (1-t)v_\theta. \tag{13}$$

During training, one uses the clean images by generating randomly corrupted images with different levels of noise, governed by the parameter $t$, and uses the network $v_\theta$ to recover the velocity $u - z$. The loss for a given triplet, $(u, z, t)$, is computed by the integrand of Equation 11. The network weights $\theta$ are trained using the AdamW (Kingma and Ba, 2017; Loshchilov and Hutter, 2019) optimizer.

**Background on DiTs.** Diffusion Transformer networks use attention-based blocks in the velocity network $v_\theta$. To this end, the image is "patchified", i.e., it is separated into patches, where each patch is comprised of $k \times k$ pixels. Given an image of shape $C \times N \times N$ where $C$ is the number of channels and $N$ is the number of pixels in each axis, each patch is flattened to a $C \cdot k \cdot k$ vector, which is considered as the feature vector that is associated with the patch. DiTs uses a fully-connected layer on the patches, yielding a coarse image of $N_{patch}^2$ patches with $C_{hid}$ channels, where $N_{patch} = N/k$ (assuming that $N_{patch}$ is an integer). The image is then flattened to generate a 2D tensor with dimensions $C_{hid} \times N_{patch}^2$. An attention layer is then used between the patches. Overall, a standard DiT block takes the computational form of:

$$
\begin{align}
h_{j+\frac{1}{2}} &= h_j + \text{ATT}(h_j), \tag{14} \\
h_{j+1} &= h_{j+\frac{1}{2}} + \text{MLP}(h_{j+\frac{1}{2}}), \tag{15}
\end{align}
$$

where ATT is an attention layer (Vaswani et al., 2017), and MLP is a standard multilayer-perceptron. Note that the MLP operates on all patches individually (similar to a dilated convolution), while the attention models generate interactions between patches.

**Patch size and complexity.** The patch size $k$ in DiTs critically controls the computation. Large $k$ inflates the MLP—whose dense layers scale as $C_{\text{hid}} \times C \times k \times k$—but reduces the number of patches to $N_{\text{patch}}^2 = N^2/k^2$, so attention couples fewer tokens; since attention cost grows quadratically in the number of patches, this yields a scaling proportional to $(N_{\text{patch}}^2 \times N_{\text{patch}}^2) = N_{\text{patch}}^4$. Conversely, a small $k$ shrinks the MLP yet explodes the patch count, making attention dominant. Empirically, a smaller $k$ improves generation quality Peebles and Xie (2023), further emphasizing attention as the bottleneck. Our PDE-SSM variant keeps the DiT block of Equation 14 unchanged except for swapping ATT with a PDE-SSM block. By the analysis in Section 2.3, this yields a complexity of $\mathcal{O}\left(N_{\text{patch}}^2 \log N_{\text{patch}}\right)$, allowing us to choose small $k$ without incurring prohibitive attention cost.

## 4 EXPERIMENTS

We evaluate *PDE-SSM-DiT* by replacing DiT attention with a spatiotemporal state-space module, holding all other components, training schedules, and sampling fixed for a clean ablation. We state the guiding research questions and evaluation protocol, then report results on four benchmarks. We analyze training dynamics via a lightweight FID proxy and study scalability under matched compute as model size and resolution increase. All experiments ran on an NVIDIA RTX6000 Ada GPU. Datasets and hyperparameters are in Appendix B.

**Research Questions.** Our experiments seek to address the following:

- **RQ1** (*PDE-SSM vs. Attention*): At fixed training objective, schedule, and sampler, does PDE-SSM-DiT match or improve the generative quality of attention-based DiT?

- **RQ2** (*Plug-and-play replacement*): When swapping only the block type (attention $\rightarrow$ PDE-SSM) in an off-the-shelf DiT, do we retain performance without retuning hyperparameters?

- **RQ3** (*Scaling and efficiency*): How do wall-clock and asymptotic complexity evolve with image and patch resolution compared to attention?

- **RQ4** (*Domain robustness*): Are the gains consistent across class-diverse (CIFAR-10, ImageNet64) and structure- or texture-centric datasets (Oxford-Flowers, LSUN-Churches)?

**Evaluation Protocol.** We conduct all experiments in the official DiT codebase (`https://github.com/facebookresearch/DiT`). We only replace the attention block from Equation 14 with our PDE-SSM block defined in Equation 2. All remaining components are held fixed,

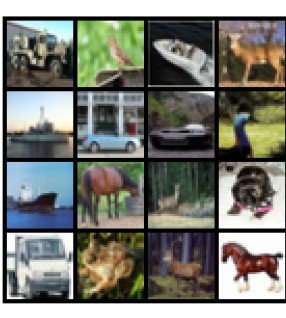 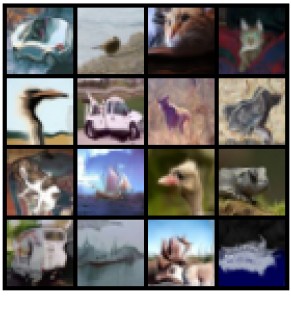 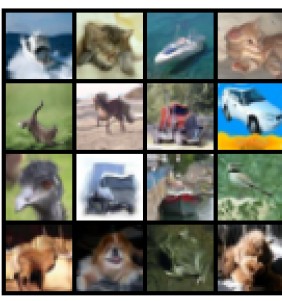

|           (a) Real images           |           (b) DiT            |         (c) PDE-SSM-DiT         |

Figure 2: CIFAR-10 Images: **(a)** real images; **(b)** DiT; **(c)** PDE-SSM-DiT. Visual quality is comparable, in congruence with Table 1.

including normalization layers, positional embeddings, loss definitions, training schedules, time discretization, and the sampling procedure. We report results for three systems: DiT (unmodified DiT with attention), UNet (a standard diffusion baseline), and PDE-SSM-DiT (DiT where attention is replaced by PDE-SSM). Because end-to-end diffusion and flow-matching pipelines vary across prior work in the training domain (pixel vs. latent) (Rombach et al., 2022), time grids (Jolicoeur-Martineau et al., 2023), learning objectives (Tong et al., 2023; Song et al., 2023), guidance mechanisms (Ho and Salimans, 2022; Dhariwal and Nichol, 2021), and samplers (Lu et al., 2022; Esser et al., 2024), our protocol yields a direct, plug-and-play comparison that isolates the architectural swap under matched settings. Together with the results reported below, this controlled evaluation highlights the effectiveness of PDE-SSM while minimizing confounding changes in training or inference.

**Training and sampling.** Unless stated otherwise, we use the default DiT hyperparameters and the standard sampling integrators from the public release cited above. Although these settings were tuned for the original DiT, we keep optimization schedules and data augmentations unchanged so that differences isolate the effect of the block replacement, directly addressing **RQ2**. We report mean metrics over three random seeds when applicable.

**Metrics.** We report Fréchet Inception Distance (FID) and, where relevant, mean-squared-error (MSE).

**Datasets.** We utilize the CIFAR-10 (Krizhevsky et al., 2009), CelebA-HQ64 (Karras et al., 2017), ImageNet64 (Deng et al., 2009), LSUN-Churches (Yu et al., 2015), and Oxford-Flowers (Nilsback and Zisserman, 2008). CIFAR-10 and ImageNet64 stress class diversity; CelebA-HQ64, Oxford-Flowers, and LSUN-Churches emphasize fine structure and global coherence, addressing **RQ4**. We provide additional details regarding the datasets and experimental settings in Appendix B.

## 4.1 IMAGE GENERATION WITH PDE-SSM-DIT

**CIFAR-10.** CIFAR-10 is a compact and diverse benchmark. We compare a small DiT ($\sim$30–34M parameters), a U-Net of comparable size, and PDE-SSM-DiT with a matched parameter budget. Results in Table 1 show parity in fidelity, while qualitative samples in Figure 2 further confirm that the block swap preserves generative quality while

| Method | #Params (M) | MSE↓ | FID (50K)↓ |
|---|---|---|---|
| DiT | 29.6 | $3.82 \times 10^{-2}$ | 4.25 |
| U-Net | 32.6 | $3.78 \times 10^{-2}$ | 4.19 |
| PDE-SSM-DiT (Ours) | 34.2 | $\mathbf{3.76 \times 10^{-2}}$ | **4.18** |

Table 1: PDE-SSM-DiT matches attention-based DiT and U-Net performance on CIFAR-10, indicating *quality parity at equal settings*, isolating the effect of the block swap (**RQ1, RQ2**).

unlocking the scaling benefits analyzed in Section 4.2 (**RQ1, RQ2**).

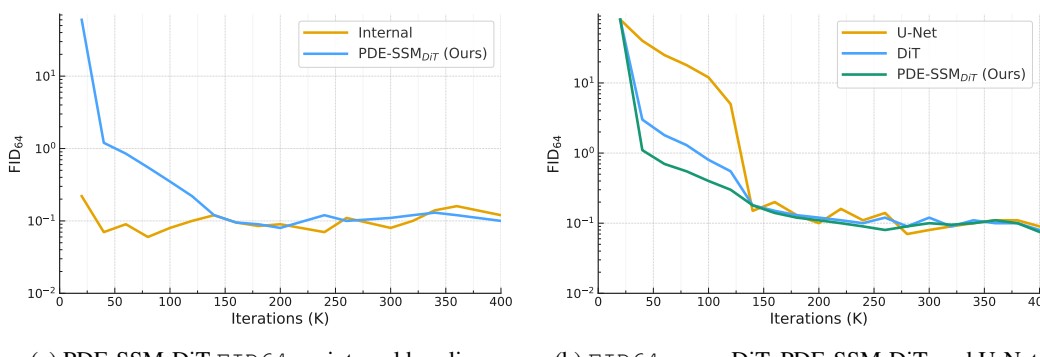

(a) PDE-SSM-DiT `FID64` vs. internal baseline.   (b) `FID64` across DiT, PDE-SSM-DiT, and U-Net.

Figure 3: ImageNet64 training. (a) All methods converge at a similar rate and to an FID score that is similar. (b) The achieved FID score is consistent with the internal FID score.

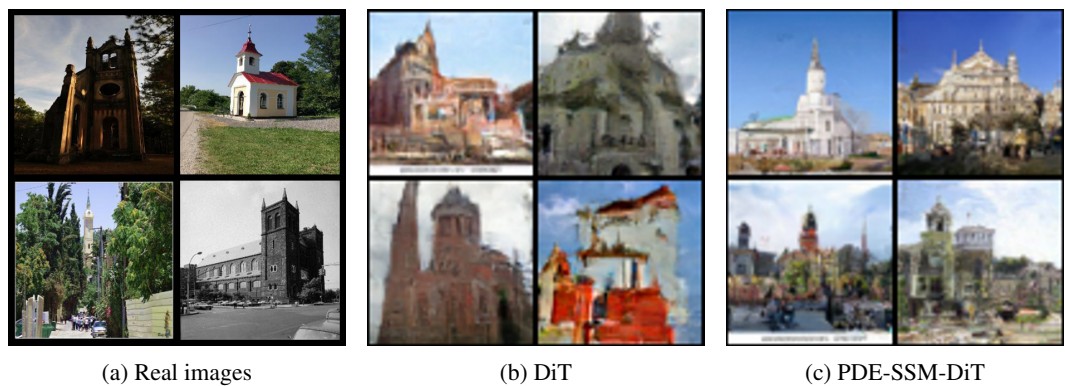

(a) Real images                (b) DiT                (c) PDE-SSM-DiT

Figure 4: LSUN-Churches generations: **(a)** real images; **(b)** DiT; **(c)** PDE-SSM-DiT.

**ImageNet**64**.** On ImageNet64, PDE-SSM-DiT achieves a modest yet consistent improvement over DiT under identical training and sampling conditions, as reported in Table 2. The model also remains competitive with U-Net at comparable parameter counts, while being strictly plug-and-play (**RQ1, RQ2, RQ4**). Literature-reported results are provided for reference, though they may differ in experimental setup and sample counts.

| Method | Params (M) | FID↓ |
|---|---|---|
| DiT | 33 | 22.9 |
| U-Net | 32 | 22.2 |
| PDE-SSM-DiT (Ours) | **31** | **22.1** |

Table 2: Unconditional Generation on ImageNet-$64 \times 64$.

Figure 3 illustrates that DiT and PDE-SSM-DiT exhibit similar convergence behavior (panel b) and reach FID scores comparable to the dataset's own internal FID (panel a). Here, internal FID refers to the baseline obtained by comparing the dataset against itself, serving as a reference for the best achievable score.

**CelebA-HQ**64**,** **LSUN-Churches** and **Oxford-Flowers:** On structure- and texture-centric datasets, PDE-SSM-DiT improves FID relative to DiT, indicating benefits for long-range coherence with small patches (**RQ1, RQ4**). Samples are shown in Figure 4, and Figure 5, while FID scores are reported in Table 3.

| Method | CelebA HQ64 | LSUN Churches | Oxford Flowers |
|---|---|---|---|
| DiT | 9.17 | 55.04 | 61.50 |
| U-Net | 11.06 | 51.56 | 57.20 |
| PDE-SSM-DiT (Ours) | **7.01** | **40.71** | **49.56** |

Table 3: FID (2048 features) on CelebA-HQ64, LSUN-Churches and Oxford-Flowers.

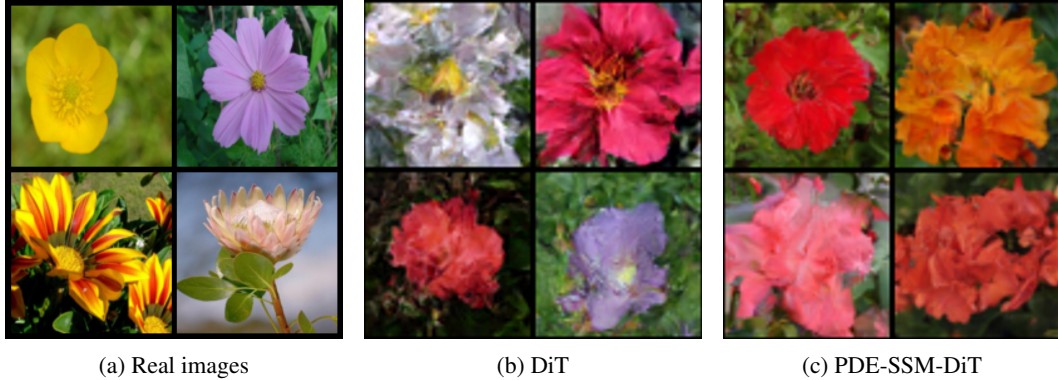

(a) Real images        (b) DiT        (c) PDE-SSM-DiT

Figure 5: Oxford-Flowers generations: **(a)** real images; **(b)** DiT; **(c)** PDE-SSM-DiT.

## 4.2 COMPUTATIONAL EFFICIENCY AND SCALING

A key benefit of PDE-SSM-DiT is favorable scaling with the number of patches. While attention scales as $O(N_{\text{patch}}^4)$ when accounting for 2D tokenization, our implementation scales as $O(N_{\text{patch}}^2 \log N_{\text{patch}})$. In practice, attention encourages large patch sizes $k \times k$ to curb quadratic attention cost, shrinking the receptive field for a fixed token budget. PDE-SSM-DiT maintains efficiency at small patches, preserving long-range dependencies without the attention bottleneck (**RQ3**). Table 4 quantifies parameter counts and per-step wall-clock across resolutions. In Appendix C we show ablation study of our model, and in Appendix E we show the spacial affect of the SSM model. Additional results for Celen-A-HQ256 are presented in Appendix D.

Table 4: DiT vs. PDE-SSM-DiT training step runtimes (in seconds). Parameters in millions (M).

| Image | Patch Size | Params (M) | | Time (s) | |
|---|---|---|---|---|---|
| | | Attn | PDE-SSM | Attn | PDE-SSM |
| 32 | 2 | 9.01 | 12.76 | $1.09 \times 10^{-1}$ | $2.14 \times 10^{-2}$ |
| 32 | 4 | 8.98 | 12.76 | $3.34 \times 10^{-2}$ | $2.40 \times 10^{-2}$ |
| 32 | 8 | 9.13 | 12.76 | $1.60 \times 10^{-2}$ | $2.40 \times 10^{-2}$ |
| 64 | 2 | 9.31 | 12.78 | $4.66 \times 10^{-1}$ | $5.58 \times 10^{-2}$ |
| 64 | 4 | 9.06 | 12.78 | $9.92 \times 10^{-2}$ | $5.77 \times 10^{-2}$ |
| 64 | 8 | 9.15 | 12.78 | $3.24 \times 10^{-2}$ | $5.84 \times 10^{-2}$ |
| 128 | 2 | 10.49 | 12.88 | $3.00 \times 10^{0}$ | $2.24 \times 10^{-1}$ |
| 128 | 4 | 9.35 | 12.88 | $4.70 \times 10^{-1}$ | $2.05 \times 10^{-1}$ |
| 128 | 8 | 9.22 | 12.88 | $1.44 \times 10^{-1}$ | $1.96 \times 10^{-1}$ |
| 256 | 2 | 15.21 | 13.27 | $3.42 \times 10^{1}$ | $8.48 \times 10^{-1}$ |
| 256 | 4 | 10.53 | 13.27 | $3.32 \times 10^{0}$ | $8.47 \times 10^{-1}$ |
| 256 | 8 | 9.52 | 13.27 | $5.07 \times 10^{-1}$ | $8.96 \times 10^{-1}$ |

**Main Takeaways.** (i) At small patches and higher resolutions, PDE-SSM-DiT yields substantial wall-clock savings vs. attention, enabling finer tokenization without exploding cost. (ii) At very large patches, attention narrows the gap, but this comes at the expense of the global receptive field.

## 5 CONCLUSION AND DISCUSSION

We extend state-space models from 1D to $n^d$ and use the resulting PDE-SSM as a drop-in replacement for attention inside DiT for image generation. The key idea is to replace the 1D ODE with a space-time PDE that combines diffusion, advection, and reaction with a trainable integration horizon, producing learned nonlocal kernels whose field of view ranges from a few pixels to the entire image. Empirically, PDE-SSM-DiT matches or improves upon attention-based DiT while scaling more favorably: attention is $O(N^2)$ in tokens $N$, whereas our spectral solver is $O(N \log N)$, enabling faster training, lower memory, and larger models under fixed budgets.

A broader takeaway is that strong generators share nonlocal mixing. UNets achieve it through multiscale downsampling, attention through pairwise correlations, and SSM or PDE-SSM through dynamical evolution that couples distant states. The emerging principle is simple: nonlocality is all you need.

**Reproducibility Statement**    We have taken several steps to ensure the reproducibility of our results. All code, model architectures, training scripts, and hyperparameter settings will be made fully public upon acceptance. We carefully document dataset preprocessing, splits, and downsampling strategies in the appendix, and provide details on batch sizes, learning rates, hidden dimensions, and model depth. Evaluation metrics, such as FID, are computed using a fixed number of samples with a specified feature extractor, consistent with established practice.

**Ethics Statement**    Our work raises minimal ethical concerns, but we acknowledge several considerations. The experiments rely on publicly available datasets including CIFAR-10, ImageNet64, CelebA-HQ, LSUN-Churches, and Oxford-Flowers, all of which are widely used in the computer vision community. CelebA-HQ contains images of human faces; we follow the dataset's licensing terms and note that no additional private or sensitive data is introduced. While our contribution is primarily methodological, we encourage responsible use of our models and caution against applications that could infringe on privacy or be used for malicious purposes. We also recognize the environmental impact of large-scale training and emphasize that our method, by improving computational efficiency, may help reduce the overall cost of generative modeling.

**Usage of Large Language Models in This Work.**    Large Language Models were used in this work for several text editing suggestions. All the concepts, code development, and original writing were carried by the authors.

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

## A    ADDITIONAL RELATED WORK

We discuss additional related work that is relevant to our paper.

Agarwal et al. (2023) introduced spectral state space models that employ spectral filtering for long-range sequence dependencies, while Fu et al. (2023) proposed hardware-efficient long convolutions for sequence modeling. Both of these works remain in the sequence modeling setting, whereas our approach generalizes state space methods to multi-dimensional spatial domains via PDE operators.

In the generative modeling domain, Goel et al. (2022) explored the use of state-space models for audio generation, and Shi et al. (2023) developed a multiresolution convolutional memory approach for capturing long-range dependencies. These models highlight the applicability of SSM-like architectures to generative tasks, but they are constrained to one-dimensional or heuristic extensions of sequential structures. By contrast, PDE-SSM provides a principled extension to higher-dimensional spatial data and integrates directly into diffusion transformers.

Smith et al. (2023) introduced simplified state space layers that offer efficient implementations of SSMs and serve as alternative baselines, and Zhang et al. (2023) proposed discrete state space formulations for time series. Both approaches focus on efficiency and simplicity in the temporal domain, whereas our work emphasizes spatial generalization and non-locality through PDE formulations. Finally, Tay et al. (2021) proposed the Long Range Arena benchmark for evaluating the scalability of models on long-range dependencies. While prior work is primarily evaluated on sequence tasks, PDE-SSM demonstrates scalability on image generation benchmarks, showcasing the benefits of our PDE-based perspective for vision applications.

Recent works investigate alternatives to attention within diffusion models. Teng et al. (2024) replace attention with Mamba-based selective state space layers to accelerate high-resolution diffusion, but their formulation operates on flattened 1D token sequences. Qin et al. (2022) introduce a dichotomous refinement mechanism for segmentation, emphasizing the importance of strong spatial priors, an idea aligned with our PDE-based inductive bias. Finally, Yan et al. (2023) propose DiffuSSM, which replaces attention with state-space layers inside diffusion models; however, DiffuSSM adapts 1D SSMs to images through sequential scanning rather than modeling true multidimensional structure. In contrast, PDE-SSM provides a learnable, physically grounded PDE operator that enables global non-local mixing with an efficient closed-form spectral implementation

## B    DATASETS AND EXPERIMENTAL SETTINGS

We elaborate on our datasets and experimental settings.

**CIFAR-10**    CIFAR-10 (Krizhevsky et al., 2009) is a canonical benchmark in computer vision, composed of 60,000 color images at a resolution of $32 \times 32$ pixels. The dataset is evenly divided into 10 object categories, including animals (e.g., birds, cats, dogs) and vehicles (e.g., airplanes, automobiles, ships). With 50,000 training images and 10,000 test images.

**ImageNet64**    ImageNet64 (Deng et al., 2009) is a downsampled variant of the large-scale ImageNet dataset, where each image is resized to $64 \times 64$ pixels. It retains the rich diversity of the original ImageNet with over 1,000 categories spanning natural objects, animals, and everyday items. Despite the reduced resolution.

**CelebA-HQ64**    CelebA-HQ (Karras et al., 2017) is a high-quality refinement of the CelebA dataset. The CelebA-HQ64 subset contains 30,000 face images at $64 \times 64$ resolution, emphasizing attributes such as hair style, facial expression, and background context.

**LSUN-Churches**    The LSUN dataset (Yu et al., 2015) contains millions of large-scale images categorized into different scene types. We focus on the LSUN-Churches subset, which consists of over 125,000 images of church images captured under diverse lighting conditions, architectural styles, and viewpoints. For our experiments, we use downsampled $64 \times 64$ versions, making the dataset suitable for structure-centric generative tasks that demand global spatial coherence.

| | CIFAR-10 | ImageNet64 | CelebA-HQ64 | LSUN-Churches | Oxford-Flowers |
|---|---|---|---|---|---|
| Max iter | 400,000 | 400,000 | 400,000 | 400,000 | 400,000 |
| Learning rate | $5 \times 10^{-4}$ | $5 \times 10^{-4}$ | $5 \times 10^{-4}$ | $5 \times 10^{-4}$ | $5 \times 10^{-4}$ |
| Hidden size | 384 | 384 | 384 | 512 | 512 |
| Batch size | 64 | 64 | 64 | 32 | 32 |
| Image size | 32 | 64 | 64 | 64 | 64 |
| Layers | 12 | 12 | 12 | 12 | 12 |
| Attention heads | 6 | 6 | 6 | 6 | 6 |

Table 5: Experimental settings and hyperparameters for different experiments.

| | Baseline | Diffusion + Reaction | Diffusion + Convection | Convection | Diffusion | Reaction |
|---|---|---|---|---|---|---|
| FID | 1.108 | 1.441 | 1.015 | 1.065 | 0.711 | 0.617 |

Table 6: Ablation study on Oxford-Flowers

**Oxford-Flowers**  The Oxford 102 Flowers dataset (Nilsback and Zisserman, 2008) consists of 8,189 images across 102 flower categories, with substantial intra-class variation in scale, pose, and background. For this dataset, we report the FID score on the same number of images as the dataset size.

Our experimental setting and details are summarized in Table 5.

## C   ABLATION STUDY

We conduct an ablation study to evaluate the contribution of the diffusion, convection, and reaction terms in our PDE-SSM formulation. Each ablation variant is trained on the Oxford-Flowers dataset for 60,000 iterations, and we report the resulting FID score computed using 64 features and 256 generated samples. The results are summarized in Table 6. These comparisons highlight the relative impact of each operator on model quality.

## D   CELEBA HQ 256

We evaluate our architecture on the CelebA-HQ 256 dataset (Karras et al., 2017) and compare its performance with the DiT baseline. As shown in Table 7, our PDE-SSM-DiT model achieves better FID scores. Representative generated samples are presented in Figure 6.

## E   SSM MAPS

To better understand the behavior of our state-space layers, we visualize the SSM maps produced by our model. These maps highlight the global structures that the SSM captures across different images, demonstrating how information propagates over large spatial extents. Figure 7 shows the SSM maps alongside their corresponding input images. In both the simple synthetic example and the real LSUN-Church (Yu et al., 2015) image, the SSM map reflects the underlying object. This illustrates the strong global bias of the SSM mechanism, which aggregates information beyond local neighborhoods and responds to the broader shape and composition of the input.

| Architecture | Parameters | FID | MSE |
|---|---|---|---|
| DiT | 32.99 M | 21.08 | 8.02e-2 |
| PDE-SSM-DiT(Ours) | 26.48 M | 18.36 | 5.37e-2 |

Table 7: Training results on CelebA HQ 256

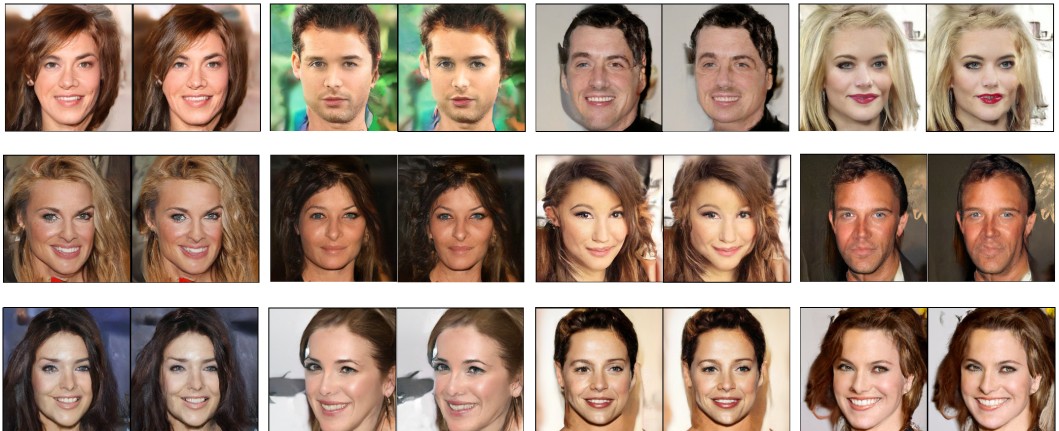

Figure 6: Sample celebrity faces generated by PDE-SSM-DiT (left) and DiT (right), trained on CelebA-HQ 256.

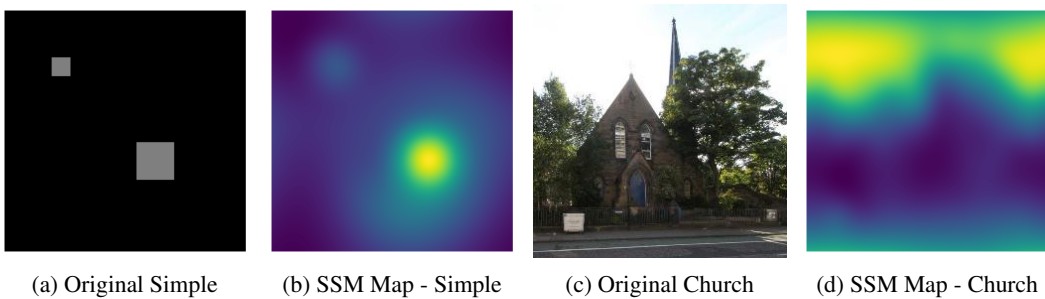

(a) Original Simple     (b) SSM Map - Simple     (c) Original Church     (d) SSM Map - Church

Figure 7: SSM maps visualization: **(a)** real simple image; **(b)** simple image SSM map; **(c)** real church image; **(d)** church SSM map.

