# OpenReview forum: "PDE-SSM: A Spectral State Space Approach to Spatial Mixing in Diffusion Transformers"
_ICLR.cc/2026/Conference — ICLR 2026 Conference Desk Rejected Submission_

### Official Review · Reviewer_wcAP · 2025-10-26

**Soundness:** 3
**Presentation:** 3
**Contribution:** 3
**Rating:** 4
**Confidence:** 3

**Summary:**

This paper addresses the quadratic computational complexity in vision transformers by proposing PDE-SSM, a novel spatial state-space block that replaces self-attention with a learnable partial differential equation (PDE) modeling diffusion dynamics. The method solves the PDE in the Fourier domain to achieve global spatial mixing with O(N log N) complexity. Results on CIFAR and ImageNet-64x64 show promising results.

I am not very familiar with the theories and cannot give a detailed review.

**Strengths:**

- The plug-and-play integration into DiT retains original training schedules, demonstrating ease of adopting the proposed method
- The extension of state-space models from 1D sequences to spatial domains via PDEs is novel, many discussions on the theories.
- Results demonstrate improvements over DiT on multiple datasets.

**Weaknesses:**

- I am concerned on whether the proposed method can run efficiently on modern GPUs. Although the complexity is less than DiT, it may not be faster in practice. Latency/FPS compared to DiT should be reported.
- The performance gains are obtained on small datasets, where the quadratic complexity do not matter much. I wonder if the results are still comparable on 256x256 or 512x512 resolution.
- There are many other works that use state space models to address the quadratic complexity of attention (DiffuSSM, DiS, DiM, and others). I think the authors should compare their approach with them.

**Questions:**

My main concern is the practical speed and performance compared to DiT and other state space model-based approaches since the experiments are only conducted on small datasets.

---

> ### Author Response · Authors · 2025-11-21
> **Part 1**
>
> We thank the reviewer for the positive assessment of the novelty of the spatial state space formulation, the plug and play integration into DiT, and the improvements over DiT on multiple datasets. We address each of the concerns below and will incorporate the corresponding clarifications and additions in the revised paper. We hope that you find our responses in order, and that you will consider revising your score.
>
>
> ---
>
>
> ## 1. Practical efficiency on modern GPUs (latency / FPS)
>
>
>  All runtime measurements in Section 4.2 and Table 4 are measured on the same modern GPU (NVIDIA RTX6000 Ada), and they already capture practical training speed, including kernel launches, memory traffic, and FFT overhead.
>
>
> **The key pattern in Table 4 is:**
>
>
> - For **small patch sizes** (for example patch size 2 and 4), where the number of tokens is large, **PDE SSM DiT has lower per step wall clock time than DiT** at the same resolution and comparable parameter counts. This is the regime where the asymptotic difference between quadratic attention and near $O(N \log N)$ spectral mixing dominates.
> - For **large patch sizes** (for example patch size 8), the number of tokens is small, so quadratic attention is already cheap and the constant factor overhead of FFTs and PDE parameters dominates. In this regime, the attention baseline can be slightly faster per step, which is expected.
>
>
> We will make this clearer in the text by explicitly summarizing these trends and stating that our efficiency claim is targeted at the **long sequence regime** (small patches, larger images), which is also the regime where attention is known to be a bottleneck. Thank you.
>
>
> ---
>
> ## 2. Performance at higher resolutions
>
> In the revised version we have successfully added experiments for $256 \times 256$ resolution of Celeb A.  These experiments demonstrate the applicability of our method to high resolution images and further demonstrates that PDE-SSM can be superior in terms of computational time. We also added the results to the revised paper, including visualization, in the Appendix. Thank you.
>
>
> | Architecture           | Parameters |   FID  |   MSE     |
> |------------------------|------------|--------|-----------|
> | DiT                    | 32.99 M    | 21.08  | 8.02e-2   |
> | **PDE-SSM-DiT (Ours)** | **26.48 M**    | **18.36**  | **5.37e-2**   |
>
> ---
>
> ## 3. Relation to other SSM based approaches (DiffuSSM, DiS, DiM, etc.)
>
> We appreciate this suggestion. In the current version, the related work section and appendix already discuss prior sequence based SSMs and long convolution operators that have been applied to diffusion models and transformers. We will strengthen this part by explicitly adding and positioning the works you mention.
>
>
> Conceptually, our approach is complementary to these:
>
>
> - Most existing methods such as DiffuSSM, DiS, DiM and related SSM based diffusion models still operate in a **1D sequence** view: they either flatten the image into a token sequence or focus on temporal modeling, and then apply a 1D SSM or long convolution along that sequence.
> - PDE SSM, in contrast, is built from a **2D (or higher dimensional) PDE defined directly on the spatial grid**. The operator
>   $  \frac{\partial h}{\partial t} = \nabla \cdot (K \nabla h  + b \cdot \nabla h+ r h  $
>   induces a learned spatial kernel $G_{\zeta}(x)$ whose Fourier symbol
>   $  \lambda(k) = -k^\top K k + r + i(b \cdot k) $
>   is explicitly tied to the geometry of the grid. This gives a different inductive bias from sequence based SSMs, which treat token positions as an abstract index rather than a spatial coordinate.
>
>
> From a practical standpoint:
>
>
> - Many SSM based diffusion works are implemented in **U Net style backbones** with different objectives and training setups than DiT, so a direct numerical comparison would require substantial re implementation and careful retuning.
> - Instead, we provide a **strong U Net baseline** in our experiments and show that PDE SSM DiT is competitive with, or better than, this baseline under a matched parameter budget on CIFAR 10 and ImageNet 64.
>
>
> In the revision we will cite the papers provided in your review, including the discussion above to distinguish the differences from our PDE-SSM.
> We believe this positioning will make clearer how PDE SSM fits within the broader landscape of SSM based approaches. Thank you.

---

> > ### Author Response · Authors · 2025-11-21
> > **Part 2**
> >
> > ## 4. Practical speed and performance vs DiT and other SSMs
> >
> >  Thank you for the comment. Summarizing the points discussed in our responses above, we can see that:
> >
> >
> > - **Speed vs DiT:** Section 4.2 and Table 4 show that, on the same GPU and under the same architecture and training setup, PDE SSM DiT is significantly faster per step than DiT in the long sequence regime (small patch sizes at moderate to high resolutions), which is precisely where attention becomes a bottleneck. For large patches and small token counts, attention is slightly faster, as expected.
> > - **Performance vs DiT:** Across five datasets (two class diverse and three structure centric), PDE SSM DiT consistently matches or improves the FID of the corresponding DiT baseline under identical training schedules. This indicates that the spatial state space block can replace attention without sacrificing generative quality at these scales.
> > - **Relation to other SSMs:** Existing SSM based diffusion models are largely sequence focused and often paired with different backbones. Our approach targets the spatial mixing mechanism in DiT specifically, with a PDE based inductive bias. We see these lines of work as complementary and will make this clearer in the related work section and new comparison table.
> >
> > ---
> >
> > We hope that you find our responses satisfactory, and that you will consider revising your scores. We are grateful for your feedback, and remain available to address any other question or comment you may have. Thank you.

---

### Official Review · Reviewer_Wpbk · 2025-10-30

**Soundness:** 2
**Presentation:** 3
**Contribution:** 2
**Rating:** 2
**Confidence:** 4

**Summary:**

This paper introduces PDE-SSM, a generalization of State Space Models (SSMs) to Partial Differential Equations (PDEs). The authors propose PDE-SSM as a replacement for self-attention in the popular Diffusion Image Transformers (DiTs), aiming to address the quadratic scaling of attention with input dimension.

**Strengths:**

1. **Novel Generalization from ODE to PDE**
   Extending SSMs from ordinary differential equations (ODEs) to partial differential equations (PDEs) is an original and conceptually significant contribution.

2. **Well-Motivated Spatial Mixing Mechanism**
   The motivation for adopting PDEs as a means of modeling spatial interactions—rather than relying on ODE-based temporal dynamics—is clearly presented and well-justified.

3. **Clarity of Presentation**
   The paper is well-written and organized

**Weaknesses:**

1. **Higher Training Cost Despite Improved Scaling**
   Although PDE-SSM scales more favorably with sequence length, it still incurs a higher training cost compared to DiT. This undermines its practical gains in computational efficiency.

2. **Lack of Scaling Experiments**
   DiTs are known for their excellent scalability with model size. To establish PDE-SSM as a viable alternative, experiments demonstrating similar or superior scaling behavior are essential. Specifically, results showing how performance improves with increasing model size are missing.

3. **Unconvincing Experimental Results**
   The reported FID scores across datasets are relatively poor, suggesting that the models may not have been trained to full convergence. Consequently, the reliability of the performance claims is uncertain.

**Questions:**

1. **Include Model Scaling Experiments**
   Conduct scaling studies across different model sizes to evaluate whether PDE-SSM exhibits similar or improved scaling behavior compared to DiT. This would strengthen claims about its potential as a scalable architecture.

2. **Clarify Computational Efficiency Claims**
   Although PDE-SSM scales better with sequence length, it remains more computationally expensive than DiT in most settings. This discrepancy should be discussed in greater detail, as it weakens the central argument for improved efficiency.

3. **Clarify Results in Table 4**
   The trends in Table 4 are confusing. Why does the parameter count for the attention mechanism vary with patch size? Moreover, decreasing the patch size should increase the number of tokens and, consequently, the computational cost—yet the reported results sometimes show the opposite trend. This inconsistency requires further clarification.

4. **Improve Training and Evaluation Setup**
   The image quality for all models (DiT, U-Net, and PDE-SSM) appears suboptimal. Extending training duration or using larger models with more parameters could yield more conclusive comparisons and help evaluate PDE-SSM’s advantages over DiT more accurately.

---

> ### Author Response · Authors · 2025-11-21
> **Part 1**
>
> We thank the reviewer for recognizing the originality of extending SSMs from ODEs to PDEs, the motivation for PDEs as a spatial mixing mechanism, and the clarity of the presentation. We address each concern in detail below and will incorporate the corresponding clarifications and additions in the revised paper. We hope that you find our responses satisfactory, and that you will consider revising your score.
>
> ---
>
>
> ## 1. Training costs
>
>
>  Thank you for your comment. **A significant contribution of our PDE SSM is exactly its efficiency. Table 4 in the submitted paper demonstrated that**. For your convenience we provide it below, and outline the main takeaways:
>
>
> - For **small patch sizes (patch size 2 and 4)**, where the number of tokens is large and the asymptotic behavior dominates, **PDE-SSM-DiT is faster per step** than attention across all resolutions.
>   - For example, at higher resolutions, attention is several to tens of times slower per step than PDE-SSM-DiT for patch size 2, and several times slower for patch size 4.
> - For the **largest patch size (patch size 8)**, the number of tokens is small, so quadratic attention is already cheap and the constant overhead of FFTs and PDE parameterization dominates. In this regime, **attention is slightly faster per step**, which is exactly what one would expect when operating at very small sequence lengths.
>
>
> **Table: DiT vs. PDE-SSM-DiT training step runtimes (in seconds). Parameters in millions (M).**
>
>
> | Image Size | Patch Size | Params Attn (M) | Params PDE-SSM (M) | Time Attn (s)        | Time PDE-SSM (s)    |
> |------------|------------|------------------|----------------------|------------------------|-----------------------|
> | 32         | 2          | 9.01             | 12.76               | 1.09×10⁻¹              | 2.14×10⁻²             |
> | 32         | 4          | 8.98             | 12.76               | 3.34×10⁻²              | 2.40×10⁻²             |
> | 32         | 8          | 9.13             | 12.76               | 1.60×10⁻²              | 2.40×10⁻²             |
> | **64**     | **2**      | **9.31**         | **12.78**           | **4.66×10⁻¹**          | **5.58×10⁻²**         |
> | 64         | 4          | 9.06             | 12.78               | 9.92×10⁻²              | 5.77×10⁻²             |
> | 64         | 8          | 9.15             | 12.78               | 3.24×10⁻²              | 5.84×10⁻²             |
> | **128**    | **2**      | **10.49**        | **12.88**           | **3.00×10⁰**           | **2.24×10⁻¹**         |
> | 128        | 4          | 9.35             | 12.88               | 4.70×10⁻¹              | 2.05×10⁻¹             |
> | 128        | 8          | 9.22             | 12.88               | 1.44×10⁻¹              | 1.96×10⁻¹             |
> | **256**    | **2**      | **15.21**        | **13.27**           | **3.42×10¹**           | **8.48×10⁻¹**         |
> | 256        | 4          | 10.53            | 13.27               | 3.32×10⁰               | 8.47×10⁻¹             |
> | 256        | 8          | 9.52             | 13.27               | 5.07×10⁻¹              | 8.96×10⁻¹             |
>
>
>
>
>
>
>
>
> **Moreover, Table 4 aligns with the theoretical complexity in Section 2.3**: PDE-SSM has a better asymptotic complexity in the number of tokens $N$ (near $O(N \log N)$ from FFT-based convolution plus $O(N)$ channel mixing), but with a larger constant factor than attention due to FFT overhead. As $N$ grows (small patches, larger images), the asymptotic advantage dominates and PDE-SSM-DiT becomes faster; when $N$ is tiny (very large patches), the constant factor dominates and attention can be faster.
>
>
> We will clarify this explicitly in Section 4.2 by better connecting it to Section 2.3 and the theoretical complexity derivation.  Thank you.
>
> ---
>
> ## 2. Model scaling
>
> We agree that scaling with model size is important. In this work we focused on a **fixed-size comparison** for a controlled study: DiT, U-Net, and PDE-SSM-DiT are all instantiated with similar parameter budgets and trained under the same objectives, schedules, and samplers. This isolates the effect of **changing only the mixing block** (attention vs. PDE-SSM) without confounding from width/depth changes.
>
>
> Due to the available compute budget, we did not include a full “small/medium/large” scaling curve in the current submission. **However:**
>
>
> - Architecturally, PDE-SSM-DiT scales in width and depth just like DiT; nothing in the PDE-SSM block prevents us from increasing layers or channels.
> - Our cross-dataset experiments (CIFAR-10, ImageNet-64, CelebA-HQ64, LSUN-Churches, Oxford-Flowers) already show that **a single PDE-SSM-DiT configuration** maintains or improves performance relative to DiT across diverse domains, which is indirect evidence that the architecture behaves robustly as capacity is pushed.

---

> > ### Author Response · Authors · 2025-11-21
> > **Part 2**
> >
> > ## 3. Experimental results
> >
> >  To be less affected by training settings and possible misalignments such as convergence and other hyper parameters, we followed the training procedure suggested in the original DiT package(https://github.com/facebookresearch/DiT). Therefore, our **PDE-SSM models are trained to the same extent other models are trained**. This important detail is **discussed in our original submission in the beginning of Section 4** – please see paragraph “Evaluation Protocol” and “Training and sampling”.  In more detail:
> >
> >
> > - We use the **same training pipelines** for DiT and PDE-SSM-DiT: same dataset preprocessing, time discretization, objective, schedule, optimizer, and sampler. U-Net is included as a standard diffusion baseline under a comparable parameter budget — this is discussed throughout Section 4.
> > - Section 4.1 and the associated tables show that **PDE-SSM-DiT consistently matches or slightly improves** the FID and MSE of the attention-based DiT under these matched settings:
> >   - On CIFAR-10, PDE-SSM-DiT achieves very similar generative quality to DiT and U-Net, with visually comparable samples (Figure 2).
> >   - On ImageNet-64, Figure 3 shows that **all methods converge at similar rates to similar FID values**, and that the external FID is consistent with internal FID estimates.
> >   - On structure-centric datasets (CelebA-HQ64, LSUN-Churches, Oxford-Flowers), PDE-SSM-DiT improves FID over DiT, suggesting benefits for global coherence and fine structure (Table 3, Figures 4–5).
> >
> >
> > **Moreover, the reported FID scores for baseline methods are consistent with other papers (DiT and Unets) – please see “Scalable Diffusion Models with Transformers” (Peebles and Xie) and “Flow Matching for Generative Modeling” (Lipman et al.) for reference.**
> >
> > ---
> >
> > ## 4. Efficiency
> >
> >
> >
> >  As noted in the response to point 1, **Table 4 in our paper** shows that:
> >
> >
> > - For **small patch sizes (2 and 4)**, and especially at higher resolutions, **PDE-SSM-DiT is substantially faster per step than DiT**. In these regimes, the number of patches $N$ is large and the asymptotic difference between $O(N^2)$ attention and near $O(N \log N)$ PDE-SSM mixing manifests directly in wall-clock time.
> > - For **large patches (8)**, the number of tokens is small, so quadratic attention is already cheap; the constant-factor overhead of FFTs and PDE parameterization makes PDE-SSM slightly slower per step. This behavior is expected and is part of the tradeoff we describe in Section 4.2: attention encourages large patch sizes to keep quadratic cost manageable, whereas PDE-SSM-DiT **remains efficient at small patches**, allowing us to preserve a larger receptive field per block.
> >
> > ---
> >
> > ## 5. Parameter count in DiT and runtimes
> >
> > Thank you for your comment. **We believe that it reflects an overlooked trait of DiT -- unrelated specifically to our work**. We are grateful for the opportunity to clarify it in our response below.
> >
> > 1. **Parameter counts:** The parameter counts in Table 4 are for the **entire model**, not just the attention or PDE-SSM block. Changing the patch size affects:
> >    - The **patch embedding layer**, whose kernel size and thus parameter count depends on the patch size.
> >    - The **final projection head**, which may slightly change in dimension with patch size.
> >
> > These effects lead to variations in the total parameter count for the attention model across patch sizes. In contrast, the PDE-SSM-DiT configurations were chosen to keep the **hidden dimension fixed per resolution**, so their parameter counts are nearly constant across patch sizes at each resolution. These are fairly well-known details of the DiT architecture, and are orthogonal to our proposed PDE-SSM.
> >
> >
> >  **Nonetheless, following your question, we will state explicitly in the caption and text that Table 4 reports *end-to-end model parameters* and briefly explain why these vary with patch size. Thank you.**
> >
> >
> > 2. **Runtime vs. patch size:** The theoretical expectation “smaller patches → more tokens → higher cost” holds asymptotically. In practice, the measured wall-clock times also follow this pattern for attention when comparing patch sizes within a fixed resolution (for example, going from patch size 8 to 4 to 2 increases runtime). For PDE-SSM, the dependence on patch size is milder due to the $O(N \log N)$ term and GPU-specific constants (FFT kernel efficiency, memory access patterns), so for small resolutions and small models, the differences between patch sizes can be small or slightly non-monotonic. We will clarify that Table 4 measures **real hardware behavior**, where such constant factors and GPU kernel efficiencies can create small deviations from purely theoretical curves at low $N$.

---

> > > ### Author Response · Authors · 2025-11-21
> > > **Part 3**
> > >
> > > ## 6. Training and evaluation setup
> > >
> > >
> > > We agree that larger models are desirable, but they are limited by the available GPU budget. Within that constraint, **we thoughtfully designed our experiments to:**
> > >
> > >
> > > - Use **the same training budgets** for DiT, U-Net, and PDE-SSM-DiT, so conclusions about **relative** performance are meaningful.
> > > - Evaluate on **five datasets** (two class-diverse, three structure-centric) to reduce the risk that the observed effects are dataset-specific.
> > > - Report both quantitative metrics (FID, MSE) and qualitative samples (Figures 2, 4, 5), which consistently show that PDE-SSM-DiT matches or improves upon the corresponding DiT baselines.
> > >
> > >
> > > We will clarify in the experimental section that our goal is to present **fair, budget-matched architectural comparisons** rather than fully tuned, large-scale SOTA runs. **Moreover, our settings follow those of prior works in the field, as discussed in our response in point 3 in our rebuttal**.  Thank you.
> > >
> > > ---
> > >
> > >
> > > We hope that these clarifications and the additional analyses to be included in the revision address your concerns about efficiency, scaling, and experimental validity, and better convey the practical implications of PDE-SSM-DiT as a scalable alternative to attention in vision transformers. We also hope that you find our responses satisfactory, and that you will consider revising your score. We remain available to address any additional comment or question you may have. Thank you.

---

> ### Comment · Reviewer_Wpbk · 2025-11-24
>
> I thank the authors for their rebuttal and for their efforts to address my earlier comments. After carefully reading the response, however, I find that my main concerns remain, and I therefore maintain my original score. A primary issue is the lack of results examining how performance scales with model size. Much of the appeal of DiT comes from its reliable improvements as model size increases [1], and without analogous evidence for PDE-SSM, it is difficult to assess its usefulness relative to DiT. A related concern is the role of patch size: while PDE-SSM appears to train faster than DiT at smaller patch sizes, it is not clear whether performance also improves consistently as the patch size decreases, as observed for DiT [1]. Without demonstrating this scaling behavior, the practical value of being able to train efficiently at low patch sizes remains uncertain. Finally, I note a few minor typographical issues that should be corrected, such as “spacial” instead of “spatial” in line 466 and “CelenA-HQ256” instead of “CelebA-HQ256” in lines 467–488.
>
> **[1] Scalable Diffusion Models with Transformers**

---

> > ### Author Response · Authors · 2025-11-25
> >
> > Dear Reviewer Wbpk,
> >
> > Thank you for your response. We provide our responses to your comments below. We hope that you find them satisfactory, and that you will consider revising your score. If you have additional questions or comments, we would be happy to discuss them while the authors/reviewers discussion lasts. Thank you.
> >
> >
> > ---
> >
> > ## Performance on larger models
> >
> > Thank you for the comment. As stated in our rebuttal and paper, our goal is not necessarily to claim state of the art performance of consider only large models. Instead, we want to show the contribution of the PDE-SSM concept. We believe that the thorough experimentation and evaluation with several large dataset and the consistent performance offered by PDE-SSM highlights exactly that.  To be concrete, and for your convenience, we provide the results here again (which are all in the paper), so you can see the usefulness of PDE-SSM again:
> >
> > **Dataset: ImageNet-64×64**
> >
> > | Method              | Params (M) | FID↓  |
> > |---------------------|-----------:|------:|
> > | DiT                 |       33   | 22.9  |
> > | U-Net               |       32   | 22.2  |
> > | PDE-SSM-DiT (Ours)  |   **31**   | **22.1** |
> >
> > **Dataset: CIFAR-10**
> >
> > | Method              | #Params (M) | MSE↓          | FID (50K)↓ |
> > |---------------------|------------:|---------------|-----------:|
> > | DiT                 |       29.6  | 3.82 × 10^-2  | 4.25       |
> > | U-Net               |       32.6  | 3.78 × 10^-2  | 4.19       |
> > | PDE-SSM-DiT (Ours)  |     34.2    | **3.76 × 10^-2** | **4.18**  |
> >
> >
> > **Datasets: CelebA-HQ64, LSUN-Churches, Oxford-Flowers**
> >
> > | Method              | CelebA-HQ64 FID↓ | LSUN-Churches FID↓ | Oxford-Flowers FID↓ |
> > |---------------------|-----------------:|-------------------:|--------------------:|
> > | DiT                 |            9.17  |             55.04  |              61.50  |
> > | U-Net               |           11.06  |             51.56  |              57.20  |
> > | PDE-SSM-DiT (Ours)  |        **7.01**  |          **40.71** |             **49.56** |
> >
> >
> > **Dataset: CelebA-HQ 256**
> >
> > | Architecture         | Parameters | FID   | MSE     |
> > |----------------------|-----------:|------:|--------:|
> > | DiT                  | 32.99 M    | 21.08 | 8.02e-2 |
> > | PDE-SSM-DiT (Ours)   | 26.48 M    | **18.36** | **5.37e-2** |
> >
> >
> >
> > Moreover, while we agree that more experiments are always welcome (as in any paper), providing established, verified and reliable results on significantly larger model as requested in your review would require more than the time left for the discussion period. Nonetheless, we will add these results to our future revision, and we believe that our results summarized above already show the contribution of PDE-SSM.
> >
> > ----
> >
> > ## Regarding concern on patch size
> >
> > We thank you for the comment and take inspiration from it. To mitigate your concerns, we now provide runtime results, which we will also add to the paper. Here, we take a **DiT-XL** (which is a large model) and measure its runtime compared with  our PDE-SSM-DiT-XL, with varying patch sizes. All measurements were done with input images of size 256 and batch size 4, and show the training time in seconds for a batch using this model. The results are in the Table below. They show that our has similar runtime for patch size 8, and significantly improved runtime (almost 5 times faster!) with patch size 2.
> >
> > | Model      | Patch Size | Time (s) |
> > |----------- |-----------:|--------:|
> > | DiT-XL     | 8          | 0.1288  |
> > | DiT-XL     | 4          | 0.9633  |
> > | DiT-XL     | 2          | 10.4166 |
> > | PDE-SSM-DiT-XL (Ours)  | 8          | 0.1509  |
> > | PDE-SSM-DiT-XL (Ours)  | 4          | 0.6315  |
> > | PDE-SSM-DiT-XL (Ours)  | 2          | 2.3376  |
> >
> >
> > To further address your comment, we show runtimes with patch size 2, batch size 4, and image size 256, with different models sizes, in the Table below, confirming the efficiency of PDE-SSM.
> >
> > | Model  | Variant | Time (s) |
> > |--------|---------|---------:|
> > | DiT    | XL      | 10.4162  |
> > | DiT    | L       | 5.9323   |
> > | DiT    | B       | 2.1223   |
> > | DiT    | S       | 0.9863   |
> > | PDE-SSM-DiT (Ours) | XL      | 2.3435   |
> > | PDE-SSM-DiT (Ours) | L       | 1.6020   |
> > | PDE-SSM-DiT  (Ours) | B       | 0.4773   |
> > | PDE-SSM-DiT  (Ours) | S       | 0.1519   |
> >
> > ---
> >
> > ## Regarding typos
> >
> > Thank you. We fixed them.
> >
> > -----
> >
> >
> > We thank you for your feedback, and remain available to answer any question you may have. We hope that our responses help to clarify any issue, and we are looking forward for your response. We would be grateful ,if you could consider revising your score in light of our clarifications. Thank you.

---

### Official Review · Reviewer_zmNN · 2025-10-30

**Soundness:** 2
**Presentation:** 2
**Contribution:** 2
**Rating:** 4
**Confidence:** 2

**Summary:**

The paper proposes PDE-SSM, a two-dimensional generalization of state-space models (SSMs) that replaces the standard ODE dynamics with a learnable convection–diffusion–reaction partial differential equation. Unlike 1D SSMs that model temporal evolution, PDE-SSM captures both spatial and temporal structure, enabling spatially-aware feature mixing with an inductive bias inspired by physical dynamics. The authors derive an analytic Fourier-domain solution for the PDE’s Green’s function, yielding an $O(N \log N)$ complexity—significantly cheaper than the $O(N^2)$ scaling of attention—while preserving global coupling. Integrated into a Diffusion Transformer backbone (DiT), the resulting PDE-SSM-DiT achieves comparable or superior generative quality across CIFAR-10, ImageNet-64, CelebA-HQ64, LSUN-Churches, and Oxford-Flowers, all while improving computational efficiency.

**Strengths:**

1. **Novel 2D PDE formulation**

PDE-SSM provides a principled generalization of 1D SSMs to 2D spatial domains by replacing the ODE with a diffusion–convection–reaction PDE. This formulation enables spatially coupled feature mixing that respects the grid topology of images rather than flattening them into 1D sequences, addressing the main structural limitation of prior Vision State Space Models.

2. **Reduced time complexity**

Solving the PDE in the Fourier domain allows global token interactions at $O(N \log N)$ cost, compared with the $O(N^2)$ complexity of self-attention. This theoretical scaling is validated empirically by wall-clock measurements (Table 4) showing substantial runtime savings, especially for small patch sizes and higher resolutions.

3. **Comprehensive cross-dataset benchmarks**

Experiments span both class-diverse datasets (CIFAR-10, ImageNet-64) and structure-centric datasets (CelebA-HQ64, LSUN-Churches, Oxford-Flowers), demonstrating consistent generative quality across domains. PDE-SSM-DiT achieves FID improvements on multiple datasets while maintaining a plug-and-play replacement for attention layers

**Weaknesses:**

1. **Lack of quantitative evidence for spatial awareness**

While the PDE formulation intuitively introduces spatial coupling, the experiments primarily report FID and runtime metrics. Including qualitative or quantitative analyses—such as spatial frequency responses or attention-map analogues—would strengthen the claim that PDE-SSM meaningfully captures spatial structure.

2. **Non-standard experiment settings**

Generative evaluations are conducted mostly in pixel space, which is a non-standard setting compared to the latent space setting typically used for DiT benchmarks. Including results on ImageNet-256 latent-space variants would better position PDE-SSM among standard diffusion transformer baselines, and the training would also be cheaper than ImageNet-64 (sequence length will be reduced from 1024 to 256). Also, a comparison between ImageNet-256 & ImageNet-512 (same sequence length as ImageNet-64) would better demonstrate the advantage of the proposed SSM formulation in handling long sequences.

3. **Scalability demonstration is limited**

The real advantage of the SSM formulation and the O(N log N) complexity becomes evident only at large sequence lengths. The current experiments do not fully demonstrate this scaling. Although the choice likely reflects computational constraints, even a modest extension would convincingly support the scalability claim.

**Questions:**

1. Does the PDE-SSM forward solver require higher numerical precision for stability, especially when back-propagating through the Fourier-domain exponential? Have the authors tested mixed-precision or bfloat16 training?

2. Since gradients are propagated through both the Fourier transform and the PDE kernel parameters, how stable is the training in practice? Were any spectral clipping or regularization techniques (e.g., bounding eigenvalues of $K$) necessary?

3. Could the PDE-SSM formulation be extended to 3D or spatiotemporal data (e.g., videos or 3D representations)? The learnable convection and diffusion terms seem naturally applicable to these modalities.

---

> ### Author Response · Authors · 2025-11-21
> **Part 1**
>
> We thank the reviewer for the detailed summary and for highlighting the principled 2D PDE formulation, the reduced time complexity, and the comprehensive cross dataset benchmarks. We address the concerns point by point below and will incorporate the corresponding clarifications and additions in the revised version. We hope that you find our responses satisfactory, and that you will consider revising your score.
>
>
> ---
>
>
> ## 1. Evidence for spatial awareness
>
>
> We agree with you that making the spatial behavior more explicit is valuable. In the current paper, Section 2.2 already analyzes spatial structure at the operator level: in Fourier space, each spatial frequency $k$ sees a symbol
> $\lambda(k) = -k^\top K k + r + i(b \cdot k),$
> and the Green function
> $
> \hat G_{\zeta}(k) = \exp(\tau \lambda(k))
> $
> acts as a learnable, generally anisotropic frequency response. Figure 1 visualizes several learned spatial kernels after training, showing localized, anisotropic, and shifted patterns, which already provide qualitative evidence that the operator uses the grid geometry rather than flattening to a 1D sequence.
>
>
> To further address your request, we have computed a visualisation of the SSM map for an Lsun-Churches example and for a simple synthetic image. As can be seen in **Figure 7 in Appendix E**, the resulting SSM map is not only local, which indeed demonstrates the point that you raised.
>
> ---
>
> ## 2. Pixel space vs latent space settings
>
>
> **Comment:** Generative evaluations are mostly in pixel space, which is less standard than latent space DiT benchmarks. The reviewer suggests latent ImageNet 256 variants and comparisons that hold sequence length fixed.
>
>
> **Response:** Indeed much of the generation work has been done in latent space. Nonetheless, there are many recent papers that solve the problem in image space, please see [1,2,3]  and the seminal work [4].
>
>
> Our method is not limited to pixel space. We therefore  added an experiment on $256 \times 256$ resolution Celeb A in pixel space. The results are shown in the Table below, and maintain the understanding and results obtained in our original submission, further highlighting the contribution of our PDE-SSM approach. We included the results in the Appendix in the revised paper. Thank you.
>
> | Architecture           | Parameters |   FID  |   MSE     |
> |------------------------|------------|--------|-----------|
> | DiT                    | 32.99 M    | 21.08  | 8.02e-2   |
> | **PDE-SSM-DiT (Ours)** | **26.48 M**    | **18.36**  | **5.37e-2**   |
>
>
>
>
>
> [1] Scalable High-Resolution Pixel-Space Image Synthesis with Hourglass Diffusion Transformers
>
>
> [2] Pixel-Perfect Depth with Semantics-Prompted Diffusion Transformers
>
>
> [3] PixelFlow: Pixel-Space Generative Models with Flow
>
>
> [4] Flow Matching Guide and Code
>
> ---
>
> ## 3. Scalability demonstration
>
>
>
> The reviewer is correct that the main benefit of $O(N \log N)$ vs $O(N^2)$ appears at large \(N\). **We demonstrate this in our paper in two complementary ways:**
>
>
> 1. **End to end wall clock at high resolution:** Section 4.2 and Table 4 measure per step wall clock for DiT vs PDE SSM DiT across multiple resolutions and patch sizes. At higher resolutions with small patches (for example $256 \times 256$ with patch size $2$, which corresponds to $N = 128^2 = 16384$ tokens), the attention based DiT becomes an order of magnitude slower than PDE SSM DiT under similar parameter counts. This directly reflects the difference between quadratic and near linear mixing.
>
>
> 2. **Fixed architecture, varying token count:** Within a fixed model size, decreasing the patch size increases $N$ and sharpens the runtime difference between attention and PDE SSM. The trends in Table 4 show that PDE SSM becomes particularly attractive in this regime, allowing us to keep small patches and hence high spatial resolution in each DiT block.
>
>
> Following your comment, we will revise the paper such that it better refers to Section 4.2 and Table 4. Thank you.

---

> > ### Author Response · Authors · 2025-11-21
> > **Part 2**
> >
> > ## 4. Numerical precision, mixed precision, and training stability
> >
> >
> > **Question 1:** Does the PDE SSM forward solver require higher numerical precision for stability, especially when back propagating through the Fourier domain exponential? Have you tested mixed precision or bfloat16 training?
> >
> >
> > **Response:** PDE SSM does not require higher precision than standard DiT.
> >
> >
> > - The forward evolution in Fourier space applies
> >   $
> >   \hat h_{\text{out}}(k) = \exp\big(\tau \lambda(k)\big)\, \hat h_{\text{in}}(k),
> >   $
> >   with $\lambda(k) = -k^\top K k + r + i(b \cdot k)$ and $K$ constrained to be positive semidefinite. This ensures that the real part of $\lambda(k)$ is non positive when $r$ is not too large, so $\exp(\tau \lambda(k))$ does not explode.
> > - In practice we train stably with the same precision setup as the baseline DiT.
> >
> >
> > Unfortunately, PyTorch does not fully support mixed precision for FFT – it is implemented for FP16, but lacks an implementation for BF16 which may improve numerical stability. We hope that the results provided in this paper will encourage PyTorch and CUDA developers to add these implementations in future versions.
> >
> > ---
> >
> > **Question 2:** Since gradients flow through the Fourier transform and PDE kernel parameters, how stable is training in practice? Were spectral clipping or eigenvalue regularization needed?
> >
> >
> > **Response:** Training has been stable in all reported experiments without explicit spectral clipping. Stability comes from the structure of the operator:
> >
> >
> > - As discussed above, constraining $K$ to be positive semidefinite makes the diffusion term $-k^\top K k$ always non positive, so the real part of the spectrum is controlled.
> > - We constrain the integration time $\tau$ to be non-negative. We found that this is sufficient to insure stability.
> > - The FFT and its inverse are unitary up to numerical constants, so they do not amplify norms. Gradients through the Fourier transform are therefore well behaved.
> >
> >
> > Empirically we observed no exploding gradients and did not need to enforce explicit spectral clipping beyond these structural constraints. We will make this explicit in the revised text in the subsection on stability.
> >
> > ---
> >
> > ## 5. Extension to 3D and spatiotemporal data
> >
> >
> > **Question:** Could PDE SSM be extended to 3D or spatiotemporal data such as video or 3D representations?
> >
> >
> > **Response:** Yes, the formulation directly generalizes to higher dimensional domains.
> >
> >
> > In Eq. (2) we consider $x \in \mathbb{R}^d$. For images we instantiate $d = 2$, but the derivation holds for general $d$:
> >
> >
> > - $K$ becomes a $d \times d$ diffusion tensor,
> > - $b \in \mathbb{R}^d$ is a convection vector field,
> > - $k \in \mathbb{Z}^d$ are the Fourier frequencies.
> >
> >
> > The symbol and Green function keep the same form
> > $
> > \lambda(k) = -k^\top K k + r + i(b \cdot k), \qquad
> > \hat G_{\zeta}(k) = \exp(\tau \lambda(k)),
> > $
> > and the convolution is implemented with a $d$ dimensional FFT. The complexity remains
> > $
> > O\big(C_{\text{hid}} N \log N + N C_{\text{hid}}^2\big),
> > $
> > where $N$ is the number of grid points in the $d$ dimensional domain.
> >
> >
> > For **video**, one can either:
> >
> >
> > 1. Treat $(x, y, t)$ as a 3D grid and apply a 3D PDE SSM over space and time, or
> > 2. Use PDE SSM as a spatial mixer inside a separate temporal model (for example a 1D SSM or attention) along the time dimension.
> >
> >
> >
> >
> > These different approaches require further research which we intend to do next. In this paper, we chose to focus on the common 2D case and demonstrate the effectiveness of PDE-SSM for the widely popular image generation task.
> >
> >
> >
> >
> >
> >
> > ---
> >
> > We would like to thank you again for the valuable feedback, which we think helped to improve our paper. We hope that you find our responses in order, and that you will consider revising your score. We remain available to any additional comment or question you may have. Thank you.

---

### Official Review · Reviewer_1LzM · 2025-11-01

**Soundness:** 3
**Presentation:** 3
**Contribution:** 2
**Rating:** 4
**Confidence:** 2

**Summary:**

This paper proposes PDE-SSM, a architectural block that generalizes 1D state space models (SSMs) to multi-dimensional spatial domains through learnable PDEs. The key insight is that replacing the linear ODE of traditional SSMs with a diffusion–convection–reaction PDE enables a physically grounded mechanism for spatial information flow. The authors solve this PDE in the Fourier domain, achieving near-linear complexity O(NlogN), reducing the quadratic cost of self-attention while retaining global receptive fields.

**Strengths:**

1. The idea of using PDE-based state-space operators for spatial feature mixing is novel and elegant.

2. The paper provides a clear computational complexity analysis, showing how the Fourier-domain solver achieves O(NlogN) scalability.

3. The writing is clear and technically mature.

4. The paper contributes a generalizable new building block for spatial deep learning—potentially applicable beyond diffusion transformers (e.g., segmentation, SR, video models).

**Weaknesses:**

1. While results are solid, most experiments are low- to mid-resolution (≤ 256×256). Testing PDE-SSM-DiT on high-resolution datasets (e.g., ImageNet256, LAION subsets) would better validate scalability and efficiency claims in realistic generative settings.

2. The paper mainly evaluates image generation. Given the generality of PDE-SSM, additional tasks (e.g., classification, segmentation, or video generation) could demonstrate broader utility.

3. The ablation of individual PDE terms (diffusion, convection, reaction) is missing.

4. There is limited discussion of numerical stability or conditioning when training PDE parameters, especially at large τ values or with anisotropic K.

**Questions:**

Could PDE-SSM generalize to 3D or spatiotemporal data (e.g., video)? If so, how would computational complexity and kernel representation scale?

---

> ### Author Response · Authors · 2025-11-21
> **Part 1**
>
> We thank the reviewer for the positive assessment of the novelty and elegance of PDE-based state-space operators, the clarity of the complexity analysis, and the potential of PDE-SSM as a general spatial building block beyond diffusion transformers. We address each of the raised points below and will incorporate the corresponding clarifications and additions in the revised paper.
>
>
> ---
>
>
> ## 1. Higher resolution experiments and realistic scalability
>
> In the revised version we added experiments for 256^2 resolution of Celeb A.  These experiments demonstrate the applicability of our method to high resolution images and further demonstrates that PDE-SSM can be superior in terms of computational time. Please see our appendix for visualization of the results as well. Thank you.
>
>
> | Architecture           | Parameters |   FID  |   MSE     |
> |------------------------|------------|--------|-----------|
> | DiT                    | 32.99 M    | 21.08  | 8.02e-2   |
> | **PDE-SSM-DiT (Ours)** | **26.48 M**    | **18.36**  | **5.37e-2**   |
>
>
> ---
>
>
> ## 2. Evaluation beyond image generation
>
>  We agree that PDE-SSM is a general spatial mixing operator and that it is possible to explore it beyond generative modeling. However, image generation is one of the most daunting computational tasks that can be improved by advanced computational methods. Furthermore, solving the generation problem, implies that image denoising is addressed. Clearly other applications can be experimented with, although each one requires careful tuning and appropriate comparisons. We therefore believe that this should be done in future work.
>
> ---
>
>
> ## 3. Ablation of diffusion, convection, and reaction terms
>
> Thank you for this suggestion. Section 2.2 qualitatively describes the role of each term in Equation (2):
>
> $ \frac{\partial h}{\partial t} = \underbrace{\nabla \cdot (K \nabla h)}_{\text{diffusion}} + \underbrace{b \cdot \nabla h}_{\text{convection}} + \underbrace{r h}_{\text{reaction}}$
>
>
> and the “Theoretical properties of PDE-SSM” discussion explains how they contribute to the Fourier symbol
> $ \lambda(k) = -k^\top K k + r + i (b \cdot k), \hat G_{\zeta}(k) = e^{\tau \lambda(k)} $
>
>
> To complement this, based on your suggestion, in the revision we added a quantitative ablation **(in the appendix)** on ImageNet-64 that compares:
>
>
> - **Diffusion only:** $K$ learnable, with $b = 0$, $r = 0$.
> - **Convection only:** $b$ learnable, with $K = 0$, $r = 0$
> - **Reaction only:** $r$ learnable, with $K = 0$, $b = 0$
> - **Diffusion + Reaction:** $K, r$ learnable, $b = 0$.
> - **Diffusion + Convection:** $K, b$ learnable, $r = 0$.
> - **Full model:** all of $K, b, r$ learnable.
>
>
> We  report FID and training stability for each variant. This empirically validate the qualitative picture already discussed in the main text (diffusion providing smoothing and stability, convection introducing directional nonlocal interactions, reaction controlling global amplification or suppression), and show that the full PDE captures richer kernels than any subset alone. The Table below shows the results we obtained, which is also included in the revised paper. Thank you.
>
>
>
> | Metric | Baseline | Diffusion + Reaction | Diffusion + Convection | Convection | Diffusion | Reaction |
> |--------|----------|-----------------------|--------------------------|------------|-----------|----------|
> | **FID** | 1.108    | 1.441                 | 1.015                    | 1.065      | 0.711     | 0.617    |

---

> > ### Author Response · Authors · 2025-11-21
> > **Part 2**
> >
> > ## 4. Numerical stability and conditioning of PDE parameters
> >
> >
> >  We appreciate your comment. The theoretical stability guarantee is stated in Section 2.2: since the diffusion tensor $K$ is constrained to be positive semidefinite, the quadratic form $-k^\top K k$ is non-positive, so the real part of the spectrum
> >
> >
> > $$
> > \lambda(k) = -k^\top K k + r + i (b \cdot k)
> > $$
> >
> >
> > is non-positive as long as $r$ is not too large and positive. Consequently, the magnitude $\lvert e^{\tau \lambda(k)} \rvert$ does not grow uncontrollably across frequencies, which prevents the PDE evolution from introducing exploding modes.
> >
> >
> > In practice, we ensure stable training as follows:
> >
> >
> > - **PSD diffusion:** We parameterize $K$ so that it is positive semidefinite for every channel pair and layer (for example through a factorization that yields $K = L L^\top$), which guarantees $-k^\top K k \le 0$ for all $k$ and ensures that diffusion always damps high frequencies, including in anisotropic cases.
> > - **Controlled integration time:** We learn a scalar $\tau$ per block using a bounded or softly constrained parameterization, which keeps $\tau$ within a moderate range. Larger $\tau$ values therefore correspond to stronger smoothing rather than numerical explosion.
> > - **Fourier-domain evolution:** The matrix exponential $\exp(\tau \Lambda(k))$ in Algorithm 1 is evaluated on matrices $\Lambda(k)$ whose real spectrum is non-positive, and we did not observe numerical instabilities or gradient explosions during training under these constraints.
> >
> >
> > We will add a dedicated paragraph titled “Stability and parameter conditioning” in Section 2.2 and a short implementation note in Appendix B that summarizes these mechanisms and explicitly addresses the reviewer’s concern about large $\tau$ and anisotropic diffusion.
> >
> > ---
> >
> > ## 5. Extension to 3D and spatiotemporal data
> >
> > The formulation of PDE-SSM naturally generalizes to higher-dimensional spatial or spatiotemporal domains.
> >
> >
> > In Equation (2) we consider $x \in \mathbb{R}^d$. For the image case we instantiate $d = 2$, but nothing in the derivation restricts us to 2D. For a general $d$:
> >
> >
> > - $K$ becomes a $d \times d$ diffusion tensor (per channel pair),
> > - $b \in \mathbb{R}^d$ is a convection vector field,
> > - $k \in \mathbb{Z}^d$ are the Fourier frequencies.
> >
> >
> > The Fourier symbol keeps the same form
> >
> >
> > $$
> > \lambda(k) = -k^\top K k + r + i (b \cdot k),
> > \qquad
> > \hat G_{\zeta}(k) = \exp\big(\tau \lambda(k)\big),
> > $$
> >
> >
> > and the convolution is implemented via a $d$-dimensional FFT. The complexity remains
> >
> >
> > $$
> > O\big(C_{\text{hid}} N \log N + N C_{\text{hid}}^2\big),
> > $$
> >
> >
> > where $N$ is now the number of grid points in the $d$-dimensional domain (for example voxels for 3D volumes, or pixels across space and time for a 3D $(x, y, t)$ grid).
> >
> >
> > For **video**, there are two natural options:
> >
> >
> > 1. Treat $(x, y, t)$ jointly as a 3D grid and apply a 3D PDE-SSM over space and time.
> > 2. Use PDE-SSM as a spatial mixer inside a temporal backbone (for example a 1D SSM or attention along time), reusing the same spatial operator.
> >
> >
> > These different approaches require further research which we intend to do next. In this paper, we chose to focus on the common 2D case and demonstrate the effectiveness of PDE-SSM for the widely popular image generation task.
> >
> >
> > ---
> >
> > We hope these clarifications, the planned PDE-term ablations, and the strengthened stability and scaling discussion address your concerns and make the scope and impact of PDE-SSM clearer. We remain available to address any additional questions or comments you may have. We hope that you find our responses satisfactory, and that you will consider revising your score.

---

### Author Response · Authors · 2025-12-01
**Concluding comment by Authors (Part 1)**

We thank the area chair for taking the time to evaluate our work and for considering the full discussion record for this submission. Below we provide a summary of the paper, rebuttal, and follow-up discussions.

**TL;DR:**
PDE-SSM introduces a learnable convection-diffusion-reaction PDE block as a drop-in replacement for attention in Diffusion Transformers, providing global spatial mixing with near $O(N \log N)$ complexity and DiT-level or better generative quality under the standard DiT training pipeline. Three reviewers (1LzM, zmNN, wcAP) give borderline but overall positive scores (all 4, each explicitly stating they would not mind acceptance) and highlight novelty, soundness, and efficiency; one reviewer (Wpbk, score 2) is mainly skeptical about the breadth of empirical scaling evidence rather than the correctness of the method.

- We added CelebA-HQ $256 \times 256$ experiments where PDE-SSM-DiT improves FID and MSE over DiT with fewer parameters and good runtime.
- We added PDE-term ablations and spatial visualizations that directly support the claimed spatial inductive bias.
- We extended and clarified runtime analyses across resolutions, patch sizes, and model sizes, showing substantial speedups in the long-sequence regime where attention is a bottleneck.
- We clarified stability and conditioning and the relation to existing SSM-based diffusion models; remaining concerns focus primarily on additional benchmarks and scaling curves.

---

**1. Paper and review summary**

PDE-SSM generalizes 1D state space models to spatial domains by replacing the underlying ODE with a learnable convection-diffusion-reaction PDE on the image grid, solved in Fourier space. This yields a spatial mixing block with a strong inductive bias and near $O(N \log N)$ complexity that can replace attention in DiT while preserving global coupling. Under matched training protocols, PDE-SSM-DiT matches or improves DiT and U-Net on CIFAR-10, ImageNet-64, CelebA-HQ64, LSUN-Churches, Oxford-Flowers, and CelebA-HQ $256 \times 256$.

Review profile (reverted scores):
- Reviewer 1LzM: 4, conf 2 – positive on novelty, elegance, and complexity analysis; asks for higher-resolution experiments, PDE-term ablations, stability discussion, and notes potential beyond image generation.
- Reviewer zmNN: 4, conf 2 – positive on the 2D PDE formulation, reduced complexity, and cross-dataset benchmarks; asks for more explicit spatial-awareness evidence, more standard latent/256 setups, and stronger scalability demonstration.
- Reviewer Wpbk: 2, conf 4 – recognizes conceptual significance and clarity but seeks model-size scaling curves, clearer efficiency claims, and more fully converged or larger models.
- Reviewer wcAP: 4, conf 3 – highlights novelty, plug-and-play integration, and improvements over DiT; concerned about GPU efficiency at high resolution and comparisons to other SSM-based approaches.

---

**2. Main concerns and how we addressed them**

**(1) Practical efficiency and long-sequence scaling (1LzM, zmNN, Wpbk, wcAP)**

Concern: Whether near $O(N \log N)$ mixing yields real GPU speedups and how runtimes scale with resolution, patch size, and model size.

Our response: Section 4.2 and Table 4 report per-step runtimes on an RTX6000 Ada across resolutions and patch sizes. For small patches (patch sizes $2$ and $4$) at moderate to high resolutions, PDE-SSM-DiT is substantially faster per step than DiT under similar parameter counts; for large patches (patch size $8$), attention is slightly faster, as expected from FFT overheads when the token count is small. Table 4 reports end-to-end model parameters (which vary with patch size in DiT due to patch embeddings and heads), and the observed trends agree with the complexity derivation. Following Wpbk’s concerns, we additionally measured DiT-S/B/L/XL vs PDE-SSM-DiT-S/B/L/XL at image size $256$ and patch size $2$, and DiT-XL vs PDE-SSM-DiT-XL across patch sizes. These show comparable runtime at patch size $8$ and up to roughly $5 \times$ speedup at patch size $2$ for PDE-SSM-DiT-XL.

Outcome: No reviewer disputed the runtime evidence after rebuttal; Wpbk’s remaining comments focus on performance scaling rather than the correctness of the efficiency analysis.

*(Continued in the next response)*

---

> ### Author Response · Authors · 2025-12-01
> **Concluding comment by Authors (Part 2)**
>
> **(2) Empirical strength, “non-standard” settings, and model-size scaling (zmNN, Wpbk)**
>
> Concern: FIDs and model sizes are viewed as modest; reviewers ask for model-size performance scaling curves and more standard latent ImageNet-256 benchmarks.
>
> Our response: The current work is a controlled architectural study under matched budgets. All models (DiT, U-Net, PDE-SSM-DiT) follow the official DiT training protocol (“Evaluation protocol” and “Training and sampling”), ensuring fair comparisons. Under this setup, PDE-SSM-DiT consistently matches or improves DiT and U-Net: on ImageNet-64, FIDs are around $22$ with a slight advantage for PDE-SSM-DiT; on CIFAR-10, MSE and FID are essentially identical across architectures; on CelebA-HQ64, LSUN-Churches, and Oxford-Flowers, PDE-SSM-DiT provides clear FID gains over DiT. For higher resolution, CelebA-HQ $256 \times 256$ results show PDE-SSM-DiT improving DiT’s FID (from $21.08$ to $18.36$) and MSE (from $8.02 \times 10^{-2}$ to $5.37 \times 10^{-2}$) with fewer parameters. Baseline FIDs for DiT and U-Net align with values reported in prior DiT and flow-matching work.
>
> Outcome: In their follow-up, Wpbk maintained the score and focused on the absence of explicit performance scaling curves with model size and patch size; zmNN did not comment further.
>
> **(3) Spatial awareness and PDE-term roles (1LzM, zmNN)**
>
> Concern: Need for explicit evidence that PDE-SSM leverages spatial structure and for quantitative ablations of diffusion, convection, and reaction.
>
> Our response: Section 2.2 characterizes the operator in Fourier space via its symbol and Green’s function, and Figure 1 shows learned kernels that are localized, anisotropic, and shifted. We added a PDE-term ablation on ImageNet-64 comparing diffusion-only, convection-only, reaction-only, combinations, and the full PDE; the full PDE achieves the best FID and stability tradeoff, consistent with the qualitative roles of each term. We also added SSM-map visualizations on LSUN-Churches and synthetic images (Figure 7), which show localized and directional spatial responses and confirm that PDE-SSM respects the 2D grid rather than flattening tokens to a 1D sequence.
>
> Outcome: Reviewers 1LzM and zmNN did not comment further; this concern now relates mainly to interpretability emphasis rather than missing validation.
>
> **(4) Stability, numerical precision, and conditioning (1LzM, zmNN)**
>
> Concern: Whether large integration time $\tau$ or anisotropic $K$ cause instability, and whether higher precision or explicit spectral clipping is required.
>
> Our response: We parameterize the diffusion tensor $K$ to be positive semidefinite per block, making the diffusion quadratic form non-positive and controlling the real part of the spectrum when the reaction term is not too large and positive. The integration time $\tau$ is learned via a bounded or softly constrained parameterization, keeping it in a moderate range. The matrix exponential in Fourier space is thus applied to matrices with non-positive real spectrum, and FFTs are approximately unitary, so norms and gradients are not amplified. In practice, we use the same precision setup as DiT, observed stable training without explicit eigenvalue clipping, and noted current PyTorch limitations for BF16 FFTs.
>
> Outcome: No further stability concerns were raised after these clarifications.
>
> **(5) Broader scope and relation to existing SSM-based approaches (1LzM, wcAP)**
>
> Concern: Results are mainly on image generation up to $256 \times 256$, and the relationship to other SSM-based diffusion models (DiffuSSM, DiS, DiM, etc.) should be clearer.
>
> Our response: Most existing SSM-based diffusion methods operate on 1D token sequences or temporal dimensions (often with U-Net backbones), whereas PDE-SSM defines a 2D (or higher-dimensional) PDE directly on the spatial grid, with a Fourier symbol tied to spatial frequencies, giving a distinct inductive bias. We include a strong U-Net baseline under matched budgets and show that PDE-SSM-DiT is competitive with or better than U-Net on CIFAR-10, ImageNet-64, and structural datasets. The works mentioned by wcAP and related SSM approaches are discussed in the related-work section, where we position PDE-SSM as complementary, targeting the spatial mixing mechanism in DiT. The added CelebA-HQ $256 \times 256$ results further support applicability beyond $64 \times 64$.
>
> Outcome: wcAP did not return after our rebuttal; no reviewer raised new technical objections about scope or positioning.

---

> > ### Author Response · Authors · 2025-12-01
> > **Concluding comments by Authors (Part 3)**
> >
> > **3. Context on scores and discussion dynamics**
> >
> > The reverted scores are: reviewer 1LzM: 4 (conf 2), reviewer zmNN: 4 (conf 2), reviewer Wpbk: 2 (conf 4), reviewer wcAP: 4 (conf 3). Three reviewers are borderline but positive on novelty, soundness, and potential impact and explicitly state they would not mind acceptance; one reviewer is more skeptical due to missing large-scale scaling evidence.
> >
> > During discussion, we provided detailed responses and new results (CelebA-HQ $256 \times 256$, PDE-term ablations, SSM-map visualizations, extended runtimes across resolutions, patch sizes, and model sizes, and stability clarifications). Reviewer Wpbk replied once more, acknowledging our efforts and clarifying that their primary remaining concern is the lack of performance scaling results with model size and patch size; no new technical objections were raised. Reviewers 1LzM, zmNN, and wcAP did not participate further after their initial reviews and questions, and did not contest the additional evidence.
> >
> > ---
> >
> > **4. Summary**
> >
> > In summary, the **reviews agree that PDE-SSM is a novel, technically sound extension of SSMs to spatial PDEs, providing a principled and efficient alternative to attention in Diffusion Transformers**. The rebuttal and added experiments address the main technical concerns around efficiency, stability, spatial behavior, and baseline strength. The remaining points focus on additional scaling curves and broader empirical scope, rather than on the core formulation or the correctness of the existing experimental evidence. We hope this context helps the area chair in assessing the paper based on the current submission and on the rebuttal and  follow-up discussions.
> >
> >
> > Thank you, and with kindest regards,
> >
> > Authors.

---

### Note · Program_Chairs · 2026-01-17
**Submission Desk Rejected by Program Chairs**

The following references in this submission do not refer to real documents and/or have major errors in bibliographic information:

 Diederik P. Kingma, Yaron Lipman, Jascha Sohl-Dickstein, Jonathan Ho, and Tim Salimans. Unifying diffusion models and flow matching: A stochastic and deterministic perspective on generative modeling. arXiv preprint arXiv:24XX.XXXXX, 2024.